# RISK-CONTROLLING MODEL SELECTION VIA GUIDED BAYESIAN OPTIMIZATION

## ABSTRACT

Adjustable hyperparameters of machine learning models typically impact various key trade-offs such as accuracy, fairness, robustness, or inference cost. Our goal in this paper is to find a configuration that adheres to user-specified limits on certain risks while being useful with respect to other conflicting metrics. We solve this by combining Bayesian Optimization (BO) with rigorous risk-controlling procedures, where our core idea is to steer BO towards an efficient testing strategy. Our BO method identifies a set of Pareto optimal configurations residing in a designated region of interest. The resulting candidates are statistically verified and the best-performing configuration is selected with guaranteed risk levels. We demonstrate the effectiveness of our approach on a range of tasks with multiple desiderata, including low error rates, equitable predictions, handling spurious correlations, managing rate and distortion in generative models, and reducing computational costs.

## 1 INTRODUCTION

Deploying machine learning models in the real-world requires balancing different performance aspects such as low error rate, equality in predictive decisions (Hardt et al., 2016; Pessach & Shmueli, 2022), robustness to spurious correlations (Sagawa et al., 2019; Yang et al., 2023), and model efficiency (Laskaridis et al., 2021; Menghani, 2023). In many cases, we can influence the model's behavior favorably via sets of hyperparameters that determine the model configuration. However, selecting such a configuration that exactly meets user-defined requirements on test data is typically non-trivial, especially when considering a large number of objectives and configurations that are costly to assess (e.g., that require retraining large neural networks for new settings).

Bayesian Optimization (BO) is widely used for efficiently selecting configurations of functions that require expensive evaluation, such as hyperparameters that govern the model architecture or influence the training procedure (Shahriari et al., 2015; Wang et al., 2022; Bischl et al., 2023). The basic concept is to substitute the costly function of interest with a cheap, and easily optimized, probabilistic surrogate model. This surrogate is used to select promising candidate configurations, while balancing exploration and exploitation. Beyond single-function optimization, BO has been extended to multiple objectives, where a set of Pareto optimal configurations that represent the best possible trade-offs is sought (Karl et al., 2022). It has also been extended to accommodate multiple inequality constraints (Gardner et al., 2014). Nevertheless, none of these mechanisms provide formal guarantees on model behavior at test time, and can suffer from unexpected fluctuations from the desired final performance (Letham et al., 2019; Feurer et al., 2023).

Addressing configuration selection from a different prospective, *Learn Then Test* (LTT) (Angelopoulos et al., 2021) is a rigorous statistical testing framework for controlling multiple risk functions with distribution-free, finite-sample validity in a model-agnostic fashion. Although providing exact theoretical verification, it becomes practically challenging to apply this framework over large configuration spaces due to increased computational costs and loss of statistical power, resulting in the inability to identify useful configurations. These challenges were addressed in the recently proposed *Pareto Testing* method (Laufer-Goldshtein et al., 2023), which combines the complementary features of multi-objective optimization and statistical testing. The core idea is that multi-objective optimization can dramatically reduce the space of configurations to consider, recovering Pareto optimal hyper-parameter combinations that are promising candidates for testing. While improving computational and statistical efficiency, the recovered subspace remains unnec-

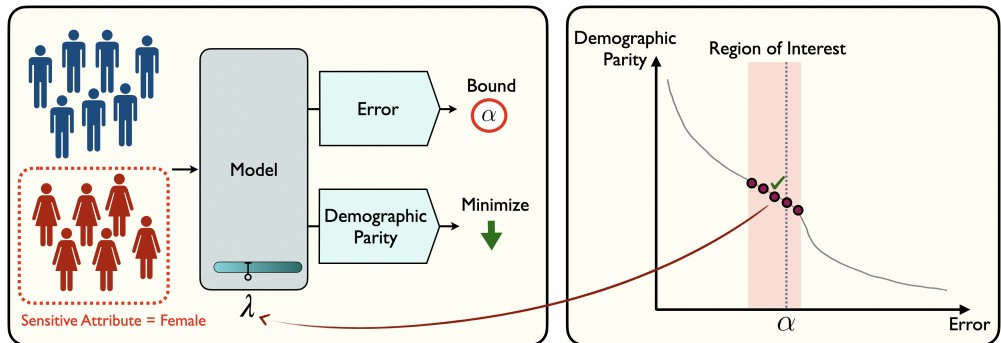

Figure 1: Demonstration of our proposed method for algorithmic fairness with gender as a sensitive attribute (left). We would like to set the model configuration $\lambda$ to minimize the difference in demographic parity, while bounding the overall prediction error by $\alpha$. Our method (right): (i) defines a region of interest in the objective space, (ii) identifies Pareto optimal solutions in this region, (iii) statistically validates the chosen solutions, and (iv) sets $\lambda$ to the best-performing verified configuration.

essarily large, containing many irrelevant configurations that are either valid but inefficient or that are highly unlikely to satisfy the constraints. Therefore, when considering expansive configuration spaces, this strategy can again become costly and statistically loose.

In this work, we propose a new synergistic approach to combine optimization and testing to achieve efficient model selection under multiple risk constraints. We introduce the notion of the *region of interest* in the objective space that is aligned with the ultimate goal of testing efficiency under limited compute budget. Our region boundaries are determined by taking into account the data sample sizes and the user-specified limits and certainty levels. Consequently, we propose an adjusted BO procedure, recovering the part of the Pareto front that intersects with the defined region of interest. The resulting focused optimization procedure recovers a dense set of configurations, representing candidates that are both effective and likely to pass the test. In the final step, we filter the chosen set by means of statistical testing to identify highly-preforming configurations that exhibit verified control.

We show that the proposed framework is flexible and can be applied in diverse contexts for both predictive and generative models, and for tuning various types of hyper-parameters that impact the model prior or post training. Specifically, we show its applicability in the domains of algorithmic fairness, robustness to spurious correlations, rate and distortion in Variational Autoencoders (VAEs), and accuracy-cost trade-offs for pruning large-scale Transformer models. See Fig. 1 for an example and a high-level illustration of the proposed method.

**Contribution.** Our main ideas and results can be summarized as follows:

1. We introduce the region of interest in the objective space that significantly limits the search space for candidate configurations in order to obtain efficient testing outcomes with less computations.
2. We define a new BO procedure to identify configurations that are Pareto optimal and lie in the defined region of interest, which are then validated via testing.
3. We present a broad range of objectives across varied tasks, where our approach can be valuable for valid control and effective optimization of diverse performance aspects, including classification fairness, predictive robustness, generation capabilities and model compression.
4. We demonstrate empirically that our proposed method selects highly efficient and verified configurations under practical budget constraints, relative to baselines.

## 2 RELATED WORK

**Conformal prediction and risk control.** Conformal prediction is a popular model-agnostic and distribution-free uncertainty estimation framework that returns prediction sets or intervals containing the true value with high probability (Vovk, 2002; Vovk et al., 2015; 2017; Lei et al., 2013; 2018; Gupta et al., 2020; Barber et al., 2021). Coverage validity, provided by standard conformal prediction, has recently been extended to controlling general statistical losses, allowing guarantees in expectation (Angelopoulos et al., 2022) or with user-defined probability (Bates et al., 2021). Our contribution builds on the foundational work by Angelopoulos et al. (2021) addressing the broader scenario of multiple risk control by selecting a proper low-dimensional hyper-parameter

configuration via multiple hypothesis testing (MHT). Additionally, we draw upon the recently introduced Pareto Testing method (Laufer-Goldshtein et al., 2023) that further improves computational and statistical efficiency by solving a multiple objective optimization (MOO) problem and focusing the testing procedure over the approximated Pareto optimal set. In this paper, we point out that recovering the entire Pareto front is redundant and costly and suggest instead to recover a focused part of the front that is aligned with the purpose of efficient testing. This enables highly-expensive hyper-parameter tuning that involves retraining of large models with a limited compute budget.

**Bayesian Optimization (BO).** BO is a commonly used sequential model-based optimization technique to efficiently find an optimal configuration for a given black-box objective function (Shahriari et al., 2015; Frazier, 2018; Wang et al., 2022). It can be applied to constrained optimization problems (Gardner et al., 2014) or multi-objective scenarios involving several conflicting objectives (Karl et al., 2022). However, when used in model hyper-paramaeter tuning, the objective functions can only be approximated through validation data, resulting in no guarantees on test time performance. To account for that we resort to statistical testing, and utilize the effectiveness of BO to efficiently explore the configuration space and identify promising candidates for testing. Closely related to our work are (Stanton et al., 2023; Salinas et al., 2023) proposing to integrate conformal prediction into BO in order to improve the optimization process under model misspecification and in the presence of observation noise. These works go in a different direction from our approach, guaranteeing coverage over the approximation of the surrogate model, while ours provides validity on configuration selection. Another recent work (Zhang et al., 2023) utilizes online conformal prediction for maintaining a safety violation rate (limiting the fraction of unsafe configurations found during BO), which differs from our provided guarantees and works under the assumption of a Gaussian observation noise.

**Multi Objective Optimization (MOO).** Simultaneously optimizing multiple black-box objective functions was traditionally performed with evolutionary algorithms, such as NSGA-II (Deb et al., 2002), SMS-EMOA (Emmerich et al., 2005) and MOEA/D (Zhang & Li, 2007). Due to the need for numerous evaluations, evolutionary methods can be costly. Alternatively, BO methods are more sample efficient and can be combined with evolutionary algorithms. Various methods were proposed exploiting different acquisition functions (Knowles, 2006; Belakaria et al., 2019; Paria et al., 2020) and selection mechanisms, encouraging diversity in the objective space (Belakaria et al., 2020) or in the design space (Konakovic Lukovic et al., 2020). The main idea behind our approach is to design a Multi-Objective-BO (MOBO) procedure that recovers a small set of configurations that are expected to be both valid and efficient, and then calibrate the chosen set via MHT (Angelopoulos et al., 2021).

Additional related work is given in Appendix A.

## 3 PROBLEM FORMULATION

Consider an input $X \in \mathcal{X}$ and an associated label $Y \in \mathcal{Y}$ drawn from a joint distribution $p_{XY} \in \mathcal{P}_{XY}$. We learn a model $f_{\boldsymbol{\lambda}} \colon \mathcal{X} \to \mathcal{Y}$, where $\boldsymbol{\lambda} \in \Lambda \subseteq \mathbb{R}^n$ is an $n$-dimensional hyper-parameter that determines the model configuration. The model weights are optimized over a training set $\mathcal{D}_{\text{train}}$ by minimizing a given loss function, while the hyper-parameter $\boldsymbol{\lambda}$ determines different aspects of the training procedure or the model final setting. For example, $\boldsymbol{\lambda}$ can weigh the different components of the training loss function, affect the data on which the model is trained, or specify the final mode of operation in a post-processing procedure.

We wish to select a model configuration $\boldsymbol{\lambda}$ according to different, often conflicting performance aspects, such as low error rate, fairness across different subpopulations and low computational costs. In many practical scenarios, we would like to constrain several of these aspects with pre-specified limits to guarantee a desirable performance in test time. Specifically, we consider a set of objective functions of the form $\ell \colon \mathcal{P}_{XY} \times \Lambda \to \mathbb{R}$. We assume that there are $c$ constrained objective functions $\ell_1, \ldots, \ell_c$, where $\ell_i(\boldsymbol{\lambda}) = \mathbb{E}_{p_{XY}}[L_i(f_{\boldsymbol{\lambda}}(X), Y, \boldsymbol{\lambda})]$ and $L_i \colon \mathcal{Y} \times \mathcal{Y} \times \Lambda \to \mathbb{R}$ is a loss function. In addition, there is a free objective function $\ell_{\text{free}}$ defining a single degree of freedom for minimization. The constraints are specified by the user and have the following form:

$$\mathbb{P}\left(\ell_i(\boldsymbol{\lambda}) \leq \alpha_i\right) \geq 1 - \delta, \quad \forall i \in \{1, \ldots, c\}, \tag{1}$$

where $\alpha_i$ is the upper bound of the $i$-th objective function, and $\delta$ is the desired confidence level. The selection is carried out based on two disjoint data subsets: (i) a validation set $\mathcal{D}_{\text{val}} = \{X_i, Y_i\}_{i=1}^{k}$ and (ii) a calibration set $\mathcal{D}_{\text{cal}} = \{X_i, Y_i\}_{i=k+1}^{k+m}$. We will use the validation data to identify a set of candidate configurations, and the calibration data to validate the identified set. Accordingly, the probabil-

ity in (1) is defined over the randomness of the calibration data, namely if $\delta = 0.1$, then the selected configuration will satisfy the constraints at least $90\%$ of the time across different calibration datasets.

We provide here a brief example of our setup in the context of algorithmic fairness and derive other applications in §6. In many cases, we wish to increase the fairness of the model without significantly sacrificing performance. For example, we would like to encourage similar true positive rates across different subpopulations, while constraining the expected error. One approach to enhance fairness involves introducing a fairness-promoting term in addition to the standard cross-entropy loss (Lohaus et al., 2020; Padh et al., 2021). In this case, $\boldsymbol{\lambda}$ represents the weights assigned to each term to determine the overall training loss. Different weights would lead to various accuracy-fairness trade-offs of the resulting model. Our goal is to select a configuration $\boldsymbol{\lambda}$ that optimizes fairness, while guaranteeing with high probability that the overall error would not exceed a certain limit.

## 4 BACKGROUND

In the following, we provide an overview on optimization of multiple objectives and on statistical testing for configuration selection, which are the key components of our method.

**Multiple Objective Optimization.** Consider an optimization problem over a vector-valued function $\boldsymbol{\ell}(\boldsymbol{\lambda}) = (\ell_1(\boldsymbol{\lambda}), \ldots, \ell_d(\boldsymbol{\lambda}))$ consisting of $d$ objectives. In the case of conflicting objectives, there is no single optimal solution that minimizes them all simultaneously. Instead, there is a set of optimal configurations representing different trade-offs of the given objectives. This is the *Pareto optimal set*, defined by:

$$\Lambda_{\mathrm{p}} = \{\boldsymbol{\lambda} \in \Lambda : \ \{\boldsymbol{\lambda}' \in \Lambda : \ \boldsymbol{\lambda}' \prec \boldsymbol{\lambda}, \boldsymbol{\lambda}' \neq \boldsymbol{\lambda} \ \} = \emptyset\}, \tag{2}$$

where $\boldsymbol{\lambda}' \prec \boldsymbol{\lambda}$ denotes that $\boldsymbol{\lambda}'$ *dominates* $\boldsymbol{\lambda}$ if for every $i \in \{1, \ldots d\}$, $\ell_i(\boldsymbol{\lambda}') \leq \ell_i(\boldsymbol{\lambda})$, and for some $i \in \{1, \ldots d\}$, $\ell_i(\boldsymbol{\lambda}') < \ell_i(\boldsymbol{\lambda})$. Accordingly, the Pareto optimal set consists of all points that are not dominated by any point in $\Lambda$. Given an approximated Pareto front $\hat{\mathcal{P}}$, a common quality measure is the hypervolume indicator (Zitzler & Thiele, 1998) defined with respect to a *reference point* $\mathbf{r} \in \mathbb{R}^d$:

$$HV(\hat{\mathcal{P}}; \ \mathbf{r}) = \int_{\mathbb{R}^d} \mathbb{1}_{H(\hat{\mathcal{P}}, \mathbf{r})} dz \tag{3}$$

where $H(\hat{\mathcal{P}}; \mathbf{r}) = \{\mathbf{z} \in \mathbb{R}^d : \exists \, \boldsymbol{p} \in \hat{\mathcal{P}} : \mathbf{p} \prec \mathbf{z} \prec \mathbf{r}\}$ and $\mathbb{1}_{H(\hat{\mathcal{P}},\mathbf{r})}$ is the Dirac delta function that equals 1 if $\mathbf{z} \in H(\hat{\mathcal{P}}; \mathbf{r})$ and 0 otherwise. An illustration is provided in Fig. E.1. The reference point defines the boundaries for the hypervolume computation. It is usually set to the nadir point that is defined by the worst objective values, so that all Pareto optimal solutions have positive hypervolume contributions (Ishibuchi et al., 2018). For example, in model compression with error and cost as objectives, the reference point can be set to $(1.0, 1.0)$, since the maximum error and the maximum normalized cost equal $1.0$. The hypervolume indicator measures both the individual contribution of each solution to the overall volume, and the global diversity, reflecting how well the solutions are distributed. It can be used to evaluate the contribution of a new point to the current approximation, defined as the Hypervolume Improvement (HVI):

$$HVI(\boldsymbol{\ell}(\boldsymbol{\lambda}), \hat{\mathcal{P}}; \mathbf{r}) = HV(\boldsymbol{\ell}(\boldsymbol{\lambda}) \cup \hat{\mathcal{P}}; \ \mathbf{r}) - HV(\hat{\mathcal{P}}; \ \mathbf{r}). \tag{4}$$

The hypervolume indicator serves both as a performance measure for comparing different algorithms and as a score for maximization in various MOO methods (Emmerich et al., 2005; 2006; Bader & Zitzler, 2011; Daulton et al., 2021).

**BO.** BO is a powerful tool for optimizing black-box objective functions that are expensive to evaluate. It uses a *surrogate model* to approximate the expensive objective function, and iteratively selects new points for evaluation based on an *acquisition function* that balances exploration and exploitation. Formally, we start with an initial pool of random configurations $\mathcal{C}_0 = \{\boldsymbol{\lambda}_0, \ldots, \boldsymbol{\lambda}_{N_0}\}$ and their associated objective values $\mathcal{L}_0 = \{\ell(\boldsymbol{\lambda}_1), \ldots, \ell(\boldsymbol{\lambda}_{N_0})\}$. Commonly, a Gaussian Process (GP) (Williams & Rasmussen, 2006) serves as a surrogate model, providing an estimate with uncertainty given by the Gaussian posterior. We assume a zero-mean GP prior $g(\boldsymbol{\lambda}) \sim \mathcal{N}(0, k(\boldsymbol{\lambda}, \boldsymbol{\lambda}))$, characterized by a kernel function $\kappa : \Lambda \times \Lambda \to \mathbb{R}$. The posterior distribution of the GP is given by $p(g|\boldsymbol{\lambda}, \mathcal{C}_n, \mathcal{L}_n) = \mathcal{N}(\mu(\boldsymbol{\lambda}), \Sigma(\boldsymbol{\lambda}, \boldsymbol{\lambda}))$, with $\mu(\boldsymbol{\lambda}) = \mathbf{k}(\mathbf{K} + \sigma^2 \mathbf{I})^{-1}\mathbf{l}$ and $\Sigma(\boldsymbol{\lambda}, \boldsymbol{\lambda}) = k(\boldsymbol{\lambda}, \boldsymbol{\lambda}) - \mathbf{k}^T (\mathbf{K} + \sigma^2 \mathbf{I})^{-1} \mathbf{k}$, where $k_i = \kappa(\boldsymbol{\lambda}, \boldsymbol{\lambda}_i)$, $K_{ij} = \kappa(\boldsymbol{\lambda}_i, \boldsymbol{\lambda}_j)$ and $l_i = \ell(\boldsymbol{\lambda}_i), i, j \in \{1, \ldots, |\mathcal{C}_n|\}$. Here $\sigma^2$ is the observation noise variance, i.e. $\ell(\boldsymbol{\lambda}_i) \sim \mathcal{N}(g(\boldsymbol{\lambda}_i), \sigma^2)$. Next, we optimize an acquisition function that

is defined on top of the surrogate model, such as probability of improvement (PI) (Kushner, 1964), expected improvement (EI) (Močkus, 1975), and lower confidence bound (LCB) (Auer, 2002). For multi-objective optimization, a GP is fitted to each objective. Then, one approach is to perform scalarization (Knowles, 2006), converting the problem back to single-objective optimization and applying one of the aforementioned acquisition functions. Another option is to use a modified acquisition function that is specified for the multi-objective case, such as expected hypervolume improvement (EHVI) (Emmerich et al., 2006) and predictive entropy search for multi-objective optimization (PESMO) (Hernández-Lobato et al., 2016). After a new configuration is selected, it is evaluated and added to the pull. This process is repeated until the maximum number of iterations is reached.

**Learn then Test (LTT) & Pareto Testing.** Angelopoulos et al. (2021) have recently proposed LTT, which is a statistical framework for configuration selection based on MHT. Given a set of constraints of the form (1), a null hypothesis is defined as $H_{\boldsymbol{\lambda}} : \exists i$ where $\ell_i(\boldsymbol{\lambda}) > \alpha_i$ i.e., that at least one of the constraints is *not* satisfied. For a given configuration, we can compute the p-value under the null-hypothesis based on the calibration data. If the p-value is lower than the significance level $\delta$, the null hypothesis is rejected and the configuration is declared to be valid. When testing multiple model configurations simultaneously, this becomes an MHT problem. In this case, it is necessary to apply a correction procedure to control the family-wise error rate (FWER), i.e. to ensure that the probability of one or more wrong rejections is bounded by $\delta$. This can become computationally demanding and result in inefficient testing when the configuration space is large. In order to mitigate these challenges, Pareto Testing was proposed (Laufer-Goldshtein et al., 2023), where the testing is focused on the most promising configurations identified using MOO. Accordingly, only Pareto optimal configurations are considered and are ranked by their approximated p-values from low to high risk. Then, Fixed Sequence Testing (FST) (Holm, 1979) is applied over the ordered set, sequentially testing the configurations with a fixed threshold $\delta$ until failing to reject for the first time. Although Pareto Testing demonstrates enhanced testing efficiency, it recovers the entire Pareto front, albeit focusing only on a small portion of it during testing. Consequently, the optimization budget is not directly utilized in a way that enhances testing efficiency, putting an emphasis on irrelevant configurations on one side and facing an excessive sparsity within the relevant area on the other.

## 5 METHOD

Our approach involves two main steps: (i) performing BO to generate a small set of potential configurations, and (ii) applying MHT over the candidate set to identify valid configurations. Considering the shortcomings of Pareto Testing, we argue that the two disjoint stages of optimization followed by testing are suboptimal, especially for resource-intensive MOO. As an alternative, we propose adjusting the optimization procedure for better testing outcomes by focusing only on the most relevant parts in the objective space. To accomplish this, we need to (i) specify a *region of interest* guided by our testing goal, and (ii) establish a BO procedure capable of effectively identifying configurations within the defined region. In the following we describe these steps in details.

### 5.1 DEFINING THE REGION OF INTEREST

We would like to define a region of interest in the objective space $\mathbb{R}^{c+1}$, where we wish to identify candidate configurations that are likely to be valid and efficient while conducting MHT. We start with the case of a single constraint ($c = 1$). Recall that in the testing stage we define the null hypothesis $H_{\boldsymbol{\lambda}} : \ell(\boldsymbol{\lambda}) > \alpha$ for a candidate configuration $\boldsymbol{\lambda}$, and compute a p-value for a given empirical loss over the calibration data $\hat{\ell}(\boldsymbol{\lambda}) = \frac{1}{m} \sum_{j=k+1}^{k+m} \ell(X_j, Y_j; \boldsymbol{\lambda})$. A valid p-value $p_{\boldsymbol{\lambda}}$ has to be super-uniform under the null hypothesis, i.e. $\mathbb{P}(p_{\boldsymbol{\lambda}} \leq u) \leq u$, for all $u \in [0, 1]$. As presented in (Angelopoulos et al., 2021), a valid p-value can be computed based on concentration inequalities that quantify how close is the sample loss to the expected population loss. When the loss is bounded by 1, we can use Hoeffding's inequality to obtain the following p-value (see Appendix E):

$$p_{\boldsymbol{\lambda}}^{\mathrm{HF}} := e^{-2m\left(\alpha - \hat{\ell}(\boldsymbol{\lambda})\right)_+^2}. \tag{5}$$

For a given significance level $\delta$, the null hypothesis is rejected (the configuration is declared to be risk-condoling), when $p_{\boldsymbol{\lambda}}^{\mathrm{HF}} < \delta$. By rearranging (5), we obtain that the maximum empirical loss

$\hat{\ell}(\boldsymbol{\lambda})$ that can pass the test with significance level $\delta$ is given by (see Appendix E):

$$\alpha^{\text{max}} = \alpha - \sqrt{\frac{\log(1/\delta)}{2m}}. \tag{6}$$

For example, consider the error rate as a loss function, which we would like to bound by $5\%$ ($\alpha = 0.05$), with significance level $\delta = 0.1$. By (6), if the empirical loss of a calibration set of size $m = 5000$ is up to $3.5\%$, then we have enough evidence to declare that this configuration is safe and its error does not exceed $5\%$.

In the BO procedure, we are interested in identifying configurations that are likely to be both valid and efficient. On the one hand, in order to be valid the loss must not exceed $\alpha^{\text{max}}$. On the other hand, from efficiency considerations, we would like to minimize the free objective as much as possible. This means that the constrained loss should be close to $\alpha^{\text{max}}$ (from bellow), since the free objective decreases as the constrained objective increases. An illustration demonstrating this idea is provided in Fig. E.2, where the irrelevant regions are: (i) the green part on the left where the configurations are not effectively minimizing $\ell_2$, and (2) the brown part on the right where the configurations are not satisfying the constraint. Ideally, we would like to find configurations with expected loss equal to the limiting testing threshold $\alpha^{\text{max}}$. However, during optimization we can only evaluate the loss on a finite-size validation data with $|\mathcal{D}_{\text{val}}| = k$. To account for that, we construct an interval $[\ell^{\text{low}}, \ell^{\text{high}}]$ around $\alpha^{\text{max}}$ based on the size of the validation data. In this region, we wish to include empirical loss values that are *likely* to correspond to an expected value of $\alpha^{\text{max}}$ based on the evidence provided by the validation data. Specifically, we consider $\hat{\ell}_1$ values that are likely to be obtained under $\ell_1 = \alpha^{\text{max}}$ with probability that is at least $\delta'$. This can be formed by defining $1 - \delta'$ confidence bounds. For example, using again Hoeffding's inequality, we obtain the following region of interest:

$$R(\alpha, k, m) = \left[ \underbrace{\alpha^{\text{max}} - \sqrt{\frac{\log(1/\delta')}{2k}}}_{\ell^{\text{low}}}, \underbrace{\alpha^{\text{max}} + \sqrt{\frac{\log(1/\delta')}{2k}}}_{\ell^{\text{high}}} \right]. \tag{7}$$

Note that setting the value of $\delta'$ is an empirical choice that is unrelated to the MHT procedure and to $\delta$. For small $\delta'$ the region expands, including more options with reduced density, while for larger $\delta'$ the region becomes smaller and denser. In any case, when $k$ increases, the width of (7) decreases as we have more confidence in the observed empirical losses of being reflective of the expected loss. In practice we use the tighter Hoeffding-Bentkus inequality for both (6) and (7) (see Appendix E).

In the case of multiple constraints, the null hypothesis is defined as $H_{\boldsymbol{\lambda}} : \exists i$ where $\ell_i(\boldsymbol{\lambda}) > \alpha_i$. A valid p-value is given by $p_{\boldsymbol{\lambda}} = \max_{i \in \{1, \dots, c\}} p_{\boldsymbol{\lambda}, i}$, where $p_{\boldsymbol{\lambda}, i}$ is the p-value corresponding to the $i$-th constraint. Consequently, we define the region of interest in the multi-constraint case as the intersection of the individual regions:

$$R(\boldsymbol{\alpha}, k, m) = \bigcap_{i=1}^{c} R(\alpha_i, k, m); \quad \boldsymbol{\alpha} = (\alpha_1, \dots, \alpha_c) \tag{8}$$

## 5.2 LOCAL HYPERVOLUME IMPROVEMENT

Given our definition of the region of interest, we derive a BO procedure that recovers Pareto optimal points in the intersection of $R(\boldsymbol{\alpha}, k, m)$ and $\mathcal{P}$. Our key idea is to use the HVI in (4) as an acquisition function and to modify it to capture only the region of interest. To this end, we properly define the reference point $\mathbf{r} \in \mathbb{R}^{c+1}$ to enclose the desired region.

Recall that the reference point defines the upper limit in each direction. Therefore, we set $r_i = \ell_i^{\text{high}}$, $i \in \{1, \dots, c\}$ using the upper bound in (7) for the constrained dimensions. We can use the maximum possible value of $\ell_{\text{free}}$ for $r_{c+1}$. However, this will unnecessarily increase the defined region, including configurations that are low-risk but do not minimize the free objective (where the constrained objectives are overly small and the free objective is overly big). Instead, we set $r_{c+1}$ to be

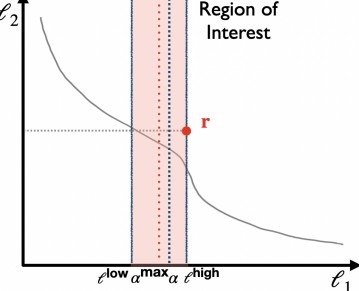

Figure 2: Proposed BO procedure for two objectives. $\ell_1$ is controlled at $\alpha$ while $\ell_2$ is minimized. The shaded area corresponds to our defined region of interest. A reference point (in red) is defined accordingly to enclose the region of interest.

the point on the free axis that correspond to the intersection of the lower limits of the constrained dimensions. For this purpose, we use the posterior mean as our objective estimator, i.e. $\hat{g} = \mu$. We define the region $R^{\text{low}} = \left\{ \boldsymbol{\lambda} : \hat{g}_1(\boldsymbol{\lambda}) < \ell_1^{\text{low}}, \ldots, \hat{g}_c(\boldsymbol{\lambda}) < \ell_c^{\text{low}} \right\}$, where the configurations are likely to be valid but inefficient. Finally, we tightly enclose this region from below in the free dimension:

$$r_{c+1} = \min_{\boldsymbol{\lambda} \in R^{\text{low}}} \hat{g}_{\text{free}}(\boldsymbol{\lambda}). \tag{9}$$

As a result, we obtain the following reference point:

$$\mathbf{r} = \left( \ell_1^{\text{high}}, \ldots, \ell_c^{\text{high}}, \min_{\boldsymbol{\lambda} \in R^{\text{low}}} \hat{g}_{\text{free}}(\boldsymbol{\lambda}) \right). \tag{10}$$

We select the next configuration by maximizing the HVI (4) with respect to this reference point:

$$\boldsymbol{\lambda}_n = \arg\max_{\boldsymbol{\lambda}} HVI(\hat{\boldsymbol{g}}(\boldsymbol{\lambda}), \hat{\mathcal{P}}; \mathbf{r}) \tag{11}$$

to recover only the relevant section and not the entire Pareto front. We evaluate the objective functions on the new selected configuration, and update our candidate set accordingly. This process of BO iterations continues until reaching the maximum budget $N$. The resulting candidate set is denoted as $\mathcal{C}^{BO}$. Our proposed BO procedure is described in Algorithm D.1 and is illustrated in Fig. 2.

Note that in MOBO it is common to use an HVI-based acquisition function that also takes into account the predictive uncertainty as in EHVI (Emmerich et al., 2005) and SMS-EGO (Ponweiser et al., 2008). However, our preliminary runs showed that these approaches do not work well in the examined scenarios with small budget ($N \in [10, 50]$), as they often generated points outside the region of interest. Similarly, for these scenarios the random scalarization approach, proposed in (Paria et al., 2020), was less effective for generating well-distributed points inside the desired region.

### 5.3 TESTING THE FINAL SELECTION

We follow (Angelopoulos et al., 2021; Laufer-Goldshtein et al., 2023) for testing the selected set. Prior to testing we filter and order the candidate set $\mathcal{C}^{BO}$. Specifically, we retain only Pareto optimal configurations from $\mathcal{C}^{BO}$, and arrange the remaining configurations by increasing p-values (approximated by $\mathcal{D}_{\text{val}}$). Next, we recompute the p-values based on $\mathcal{D}_{\text{cal}}$ and perform FST, where we start testing from the first configuration and continue until the first time the p-value exceeds $\delta$. As a result, we obtain the validated set $\mathcal{C}^{\text{valid}}$, and choose a configuration minimizing the free objective:

$$\boldsymbol{\lambda}^* = \min_{\boldsymbol{\lambda} \in \mathcal{C}^{\text{valid}}} \ell_{\text{free}}(\boldsymbol{\lambda}). \tag{12}$$

Our method is summarized in Algorithm D.2. As a consequence of (Angelopoulos et al., 2021; Laufer-Goldshtein et al., 2023) we achieve a valid risk-controlling configuration, as we now formally state.

**Theorem 5.1.** *Let $\mathcal{D}_{\text{val}} = \{X_i, Y_i\}_{i=1}^k$ and $\mathcal{D}_{\text{cal}} = \{X_i, Y_i\}_{i=k+1}^{k+m}$ be two disjoint datasets. Suppose the p-value $p_{\boldsymbol{\lambda}}$, derived from $\mathcal{D}_{\text{cal}}$, is super-uniform under $\mathcal{H}_{\boldsymbol{\lambda}}$ for all $\boldsymbol{\lambda}$. Then the output $\boldsymbol{\lambda}^*$ of Algorithm D.2 satisfies Eq. (1).*

In situations where we are unable to identify any statistically valid configuration (i.e., $\mathcal{C}^{\text{valid}} = \emptyset$), we set $\boldsymbol{\lambda} = \texttt{null}$. In practice, the user can choose limits $\alpha_1, \ldots, \alpha_c$ that are likely to be feasible based on the initial pool of configurations $\mathcal{C}_0$ that is generated at the beginning of the BO procedure. Specifically, the user may select $\alpha_i \in [\min_{\boldsymbol{\lambda} \in \mathcal{C}_0} \ell_i(\boldsymbol{\lambda}), \max_{\boldsymbol{\lambda} \in \mathcal{C}_0} \ell_i(\boldsymbol{\lambda})], i \in \{1, \ldots, c\}$, and can further refine this choice during the BO iterations as more function evaluations are accumulated.

## 6 APPLICATIONS

We demonstrate the effectiveness of our proposed method for different tasks with diverse objectives, where the definition of $\boldsymbol{\lambda}$ and its effect prior or post training, vary per setting.

**Classification Fairness.** In many classification tasks, it is important to take into account the behavior of the predictor with respect to different subpopulations. Assuming a binary classification task and a binary sensitive attribute $a = \{-1, 1\}$, we consider the Difference of Demographic Parity (DDP) as a fairness score (Wu et al., 2019):

$$\text{DDP}(f) = \mathbb{E}\left[ \mathbb{1}_{f(x) > 0} | a = -1 \right] - \mathbb{E}\left[ \mathbb{1}_{f(x) > 0} | a = 1 \right]. \tag{13}$$

We define the following loss parameterized by $\lambda$:

$$R(f;\lambda) = (1 - \lambda) \cdot \text{BCE}(f) + \lambda \cdot \widehat{\text{DDP}}(f), \tag{14}$$

where BCE is the binary cross-entropy loss, and $\widehat{\text{DDP}}$ is the hyperbolic tangent relaxation of (13) (Padh et al., 2021). Changing the value of $\lambda$ leads to different models that trade-off accuracy for fairness. In this setup, we have a 1-dimensional hyperparamter $\lambda$ and two objectives: (i) the error of the model $\ell_{\text{err}}(\lambda) = \mathbb{E}\left[\mathbb{1}_{f_\lambda(X) \neq Y}\right]$, and (ii) the DDP defined in (13) $\ell_{\text{ddp}}(\lambda) = \text{DDP}(f_\lambda)$.

**Classification Robustness.** Predictors often rely on spurious correlations found in the data (such as background features), which leads to significant performance variations among different subgroups. Recently, Izmailov et al. (2022) demonstrated that models trained using expected risk minimization surprisingly learn core features in addition to spurious ones. Accordingly, they proposed to enhance model robustness by retraining the final layer on a balanced dataset. We adapt their approach to obtain different configurations, offering a trade-off between robustness and average performance.

Given a dataset $\mathcal{D}$ (either the training set or a part of the validation set) we denote by $\mathcal{D}_b$ a balanced subset of $\mathcal{D}$ with equal number of samples per subgroup, and by $\mathcal{D}_u$ a random (unbalanced) subset of $\mathcal{D}$. We define a parameterized dataset $\mathcal{D}_\lambda$ in the following way. Let $B \sim \text{Bern}(\lambda)$ denote a Bernoulli random variable with parameter $\lambda$. We randomly draw $K$ i.i.d samples $\{B_i\}_{i=1}^K$, and construct $\mathcal{D}_\lambda = \{X_i, Y_i\}_{i=1}^K$, where $(X_i, Y_i)$ are randomly drawn from $\mathcal{D}_b$ if $B_i = 1$ or from $\mathcal{D}_u$, otherwise. We train the last layer with binary cross-entropy loss on the resulting dataset $\mathcal{D}_\lambda$. As a result, we have a 1-dimensional hyper-parameter $\lambda$ that controls the degree to which the dataset is balanced. We define two objective functions: (i) the average error $\ell_{\text{err}}(\lambda) = \mathbb{E}\left[\mathbb{1}_{f_\lambda(X) \neq Y}\right]$, and (ii) the worst error over all subgroups $\ell_{\text{worst-err}}(\lambda) = \max_{g \in \mathcal{G}} \mathbb{E}\left[\mathbb{1}_{f_\lambda(X) \neq Y} | G = g\right]$ where $G \in \mathcal{G}$ is the group label.

We also examine the case of *selective* classification and robustness. The selective classifier can abstain from making a prediction when the confidence is lower then a threshold $\tau$, i.e. $f_\lambda(x) < \tau$. In this case, we have a 2-dimensional hyper-parmeter $\boldsymbol{\lambda} = (\lambda, \tau)$ and an additional objective function of the mis-coverage rate (where the predictor decides to abstain) $\ell_{\text{mis-cover}}(\lambda) = \mathbb{E}\left[\mathbb{1}_{f_\lambda(x) < \tau}\right]$.

**VAE.** Variational Autoencoders (VAEs) (Kingma & Welling, 2013; Rezende et al., 2014) are generative models that leverage a variational approach to learn the latent variables underlying the data, and can generate new samples by sampling from the learned latent space. We focus on a $\beta$-VAE (Higgins et al., 2016), which balances the reconstruction error (distortion) and the KL divergence (rate):

$$R(f;\beta) = \mathbb{E}_{p_d(x)}\left[\mathbb{E}_{q_\phi(z|x)}\left[-\log p_\theta(x|z)\right]\right] + \beta \cdot \mathbb{E}_{p_d(x)}\left[D_{KL}(q_\phi(z|x)||p(z))\right], \tag{15}$$

where $f$ consists of an encoder $q_\phi(z|x)$ and a decoder $p_\theta(x|z)$, parameterized by $\phi$ and $\theta$, respectively, and $p(z)$ is the latent prior distribution. Generally, models with low distortion perform high-quality reconstruction but generate less realistic samples and vice versa. We have a single parameter $\lambda = \beta$ and two objectives $\ell_{\text{recon}}(f)$, $\ell_{\text{KLD}}(f)$ defined by the left and right terms in (15), respectively.

**Transformer Pruning.** We adopt the three dimensional transformer pruning scheme proposed in (Laufer-Goldshtein et al., 2023): (i) token pruning, removing unimportant tokens from the input sequence, (ii) layer early-exiting, computing part of the model's layers for easy examples, and (iii) head pruning, removing attention heads from the model architecture. We obtain $\boldsymbol{\lambda} = (\lambda_1, \lambda_2, \lambda_3)$ with the three thresholds controlling the pruning in each dimension, and consider two objectives: (i) the accuracy difference between the pruned model and the full model $\ell_{\text{diff-acc}}(\lambda) = \mathbb{E}\left[\mathbb{1}_{f(X)=Y} - \mathbb{1}_{f_{\boldsymbol{\lambda}}(X)=Y}\right]$ and (ii) the respective cost ratio $\ell_{\text{cost}}(\lambda) = \mathbb{E}\left[\frac{C(f_{\boldsymbol{\lambda}}(X))}{C(f(X))}\right]$.

# 7 EXPERIMENTS

We briefly describe the experimental setup and present our main results. Detailed setup information, as well as additional results are provided in Appendixes B and C, respectively.

**Baselines.** We compare the proposed method to other baselines that differ only in the first stage by their optimization mechanism. The second testing stage is the same for all baselines (and the proposed method), therefore all baselines can be considered as variants of Pareto Testing (Laufer-Goldshtein et al., 2023). We define two simple baselines: UNIFORM - a uniform grid in the hyper-parameter space. RANDOM - a uniform random sampling for $n = 1$ and Latin Hypercube Sampling (LHS) (McKay et al., 2000) for $n > 1$. In addition, we compare to multi-objective optimizers: HVI (same acquisition function as in the proposed method) and EHVI (Emmerich et al.,

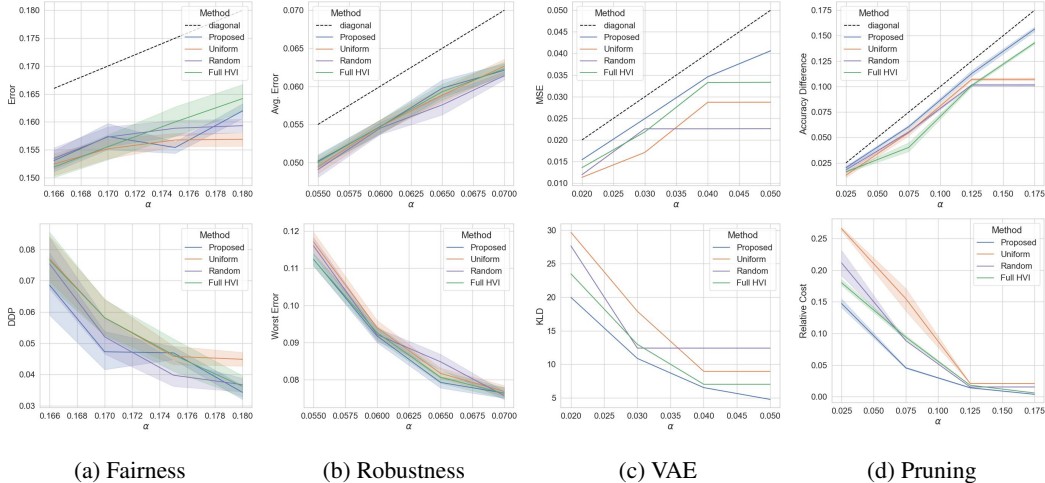

(a) Fairness · (b) Robustness · (c) VAE · (d) Pruning

Figure 3: Two objectives. Presents constrained (top) and free objectives (bottom). PAREGO and EHVI baselines appear in Fig. C.1 for the sake of clarity.

2006) with reference point defined by the maximum loss values, and PAREGO (Knowles, 2006; Cristescu & Knowles, 2015) using SMAC3 implementation (Lindauer et al., 2022). We choose the values of $\alpha$ for each task according to the range obtained from the initial pool of configurations. See table B.1 for the range values for each objective. We set $\delta = 0.1$ and $\delta' = 0.0001$.

**Datasets.** We use the following datasets: **Fairness - Adult** (Dua et al., 2017), predict if the income is above 50k$ with gender as a sensitive attribute; **Robustness - CelebA** (Lin et al., 2019), predict if a person has blond hair, where the spurious correlation is the gender; **VAE - MNIST** (LeCun, 1998); **Pruning - AG News** (Zhang et al., 2015), topic news classification.

**Two objectives.** We examine the following scenarios: **Fairness** - error is controlled and DDP is minimized; **Robustness** - avg. error is controlled and worst error is minimized; **VAE** - reconstruction error is controlled and KLD is minimized; **Pruning** - error difference is controlled and relative cost is minimized. Results are presented in Figs. 3 and C.1 , showing the mean scores over 50 random calibration and test splits. Shaded regions correspond to 95% CI. We see that the proposed method is superior over all baselines in almost all cases. The other baselines present an inconsistent behavior, showing desired performance in certain tasks or for specific $\alpha$ values, and worse performance in other cases. This is attributed to the fact that for the baselines the way the configurations are distributed over the Pareto front is arbitrary. Therefore, sometimes by chance we obtain configurations that are near the testing limit (hence efficient), while in other cases the nearest configuration is far-away (inefficient). On the contrary, the proposed method obtains a dense sampling of the relevant part of the Pareto front, which results in tighter and more stable control across different conditions.

**Additional Results.** We consider a three-objective scenario of selective classification and robustness, constraining the average error and the miscoverage rate and minimizing the worst error. We see on Figs. C.2 and C.3 that the proposed method outperforms the baselines. We also explore the budget for which we can match the performance of a dense uniform grid (with over 6K points) in Fig. C.4. We show that $N = 60$ is sufficient, highlighting the computational advantage of the proposed method. In addition, we examine the influence of $\delta'$ in Fig. C.6, showing that the method is generally insensitive to $\delta'$. Finally, Fig. C.7 shows that using the proposed region is preferable over a single-sided upper bound, implying that it is important to exclude inefficient configurations.

## 8 CONCLUSION

We present a flexible framework for reliable model selection that satisfy statistical risk constraints, while optimizing additional conflicting metrics. We define a confined region in the objective space that is a promising target for testing, and propose a BO method that identifies Pareto optimal configurations within this region. By statistically validating the candidate set via multiple hypothesis testing, we obtain verified control guarantees. Our experiments have demonstrated the effectiveness of our approach for tuning different types of hyperparameters across various tasks and objectives, including high-accuracy, fairness, robustness, generation and reconstruction quality and cost considerations.

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

## A  ADDITIONAL RELATED WORK

**Gradient-Based MOO.**  When dealing with differentiable objective functions, gradient-based MOO algorithms can be utilized. The cornerstone of these methods is Multiple-Gradient Descent (MGD) (Sener & Koltun, 2018; Désidéri, 2012), which ensures that all objectives are decreased simultaneously, leading to convergence at a Pareto optimal point. Several extensions were proposed to enable convergence to a specific point on the front defined by a preference vector (Lin et al., 2019; Mahapatra & Rajan, 2020), or learning the entire Pareto front, using a preference-conditioned model (Navon et al., 2020; Lin et al., 2020; Chen & Kwok, 2022; Ruchte & Grabocka, 2021). However, this line of research focuses on differentiable objectives, optimizing the loss space used during training, which is typically different from the ultimate non-differentiable metrics used for evaluation (e.g. error rates). Furthermore, it focuses on recovering a single or multiple (possibly infinitely many) Pareto optimal points, without addressing the actual selection of model configuration under specific constraints, which is the problem we tackle in this paper.

## B  IMPLEMENTATION AND DATASET DETAILS

We provide here further details on the datasets, model architectures, training procedures, and examined scenarios, which were used in our experiments.

**Datasets and Evaluation Details.**  Detailed information on the datasets and the examined scenarios is provided in Table B.1, including: the number of samples for each data split (train/validation/calibration/test), the best and the worst performance for each objective (over the validation set) and the optimization budget. We emphasize again the purpose of each data split. The training dataset is used for learning the model's parameters. The validation data is used for BO in Algorithm D.1 and for ordering the chosen configurations. The calibration data is used for the testing procedure. The final chosen $\boldsymbol{\lambda}^*$ (12) is used for setting the model configuration. Finally, the performance of the selected model is examined over the test dataset. We repeat the experiments, with different splits to calibration and test sets, and report the scores of the test data, averaged over different trials. We define the range of $\alpha$ bounds according to the values observed for the initial pool of configurations that is generated at the beginning of the BO procedure. The minimum and the maximum edge points obtained for each task appear in table B.1, and are also observed in the examples shown on Fig. C.5. We choose values in between these extreme edge points, but not too close to either side, since too small values may not be statistically achievable and too large values are trivially satisfied (with tighter control not significantly improving the free objective).

**Fairness.** Our model is a 3-layer feed-forward neural network with hidden dimensions $[60, 25]$. We train all models using Adam optimizer with learning rate $1e-3$ for $50$ epochs and batch size $256$.

**Robustness.** We use a ResNet-50 model pretrained on ImageNet. We train the models for $50$ epochs with SGD with a constant learning rate of $1e-3$, momentum decay of $0.9$, batch size $32$ and weight decay of $1e-4$. We use random crops and horizontal flips as data augmentation. We use half of the CelebA validation data to train the last layer, and the other half for BO.

**VAE.** We use the implementation provided by (Chadebec et al., 2022) of a ResNet-based encoder and decoder, trained using AdamW optimizer with $\beta_1 = 0.91$, $\beta_2 = 0.99$, and weight decay $0.05$. We set the learning to $1e-4$ and the batch size to $64$. The training process consisted of $10$ epochs. We use binary-cross entropy reconstruction loss for training the model, and the mean squared error normalized by the total number of pixels ($728$) as the reconstruction objective function for hyperparameter tuning.

**Pruning.** We use a BERT-base model (Devlin et al., 2018) with $12$ layers and $12$ heads per layer. We follow the recipe in (Laufer-Goldshtein et al., 2023) and attach a prediction head and a token importance predictor per layer. The core model is first finetuned on the task. We compute the attention head importance scores based on 5K held-out samples out of the training data. We freeze the backbone model and train the early-exit classifiers and the token importance predictors on the training data (115K samples).

Each prediction head is a 2-layer feed-forward neural network with $32$ dimensional hidden states, and ReLU activation. The input is the hidden representation of the [CLS] token concatenated with the hidden representation of all previous layers, following (Wołczyk et al., 2021).

Table B.1: Datasets Details

| Dataset | Train | Validation | Calibration | Test | Objectives $(\ell_1, \ell_2)$ | (best $\ell_1$, worst $\ell_2$) | (worst $\ell_1$, best $\ell_2$) | $N$ | $N_0$ |
|---|---|---|---|---|---|---|---|---|---|
| Adult | 32,559 | 3,618 | 4,522 | 4,523 | (Error, DDP) | (0.154, 0.145) | (0.225, 0.01) | 10 | 5 |
| CelebA | 162,770 | 19,867 | 9,981 | 9,981 | (Avg. Error, Worst Error) | (0.045, 0.62) | (0.089, 0.11) | 15 | 5 |
| MNIST | 50,000 | 10,000 | 5,000 | 5,000 | (Recon. Error, KLD) | (0.008, 80) | (0.072, 0.01) | 10 | 5 |
| AG News | 120,000 | 2,500 | 2,500 | 2,600 | (Acc. Difference, Rel. Cost) | (0.0, 1.0) | (0.8, 0.0) | 50 | 30 |

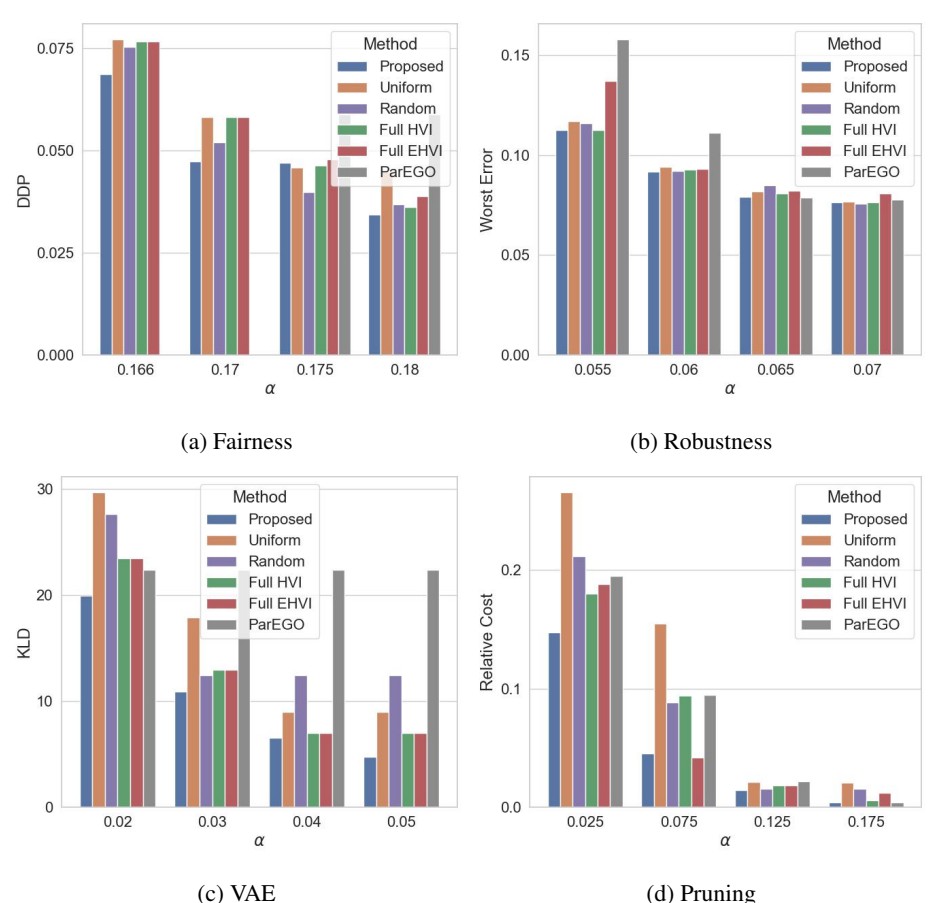

(a) Fairness

(b) Robustness

(c) VAE

(d) Pruning

Figure C.1: Two objectives, additional baselines (50 random splits) - presenting the scores obtained for the free objectives. For the fairness task, ParEGO failed to find a valid configurations in 14% and 2% of the runs for $\alpha = 0.166$ and $\alpha = 0.17$, respectively. Therefore we do not show the scores of to this baseline in these two cases.

Similarly, each token importance predictor is a 2-layer feed-forward neural network with 32 dimensional hidden states, and ReLU activation. The input is the hidden representation of each token in the current layer and all previous layers (Wołczyk et al., 2021).

## C   ADDITIONAL RESULTS

In this section, we describe additional experiments and results.

**Varying Optimization Budget.** We examine the effect of varying the optimization budget $N$. We show results for the pruning task with $N \in \{10, 20, 50\}$. In addition, we compare to a dense grid with uniform sampling of all 3 hyperparmeters with a total of $N = 6480$ configurations. We see on Fig. C.4 that the relative cost gradually improves with the increase in $N$ and reaches the performance

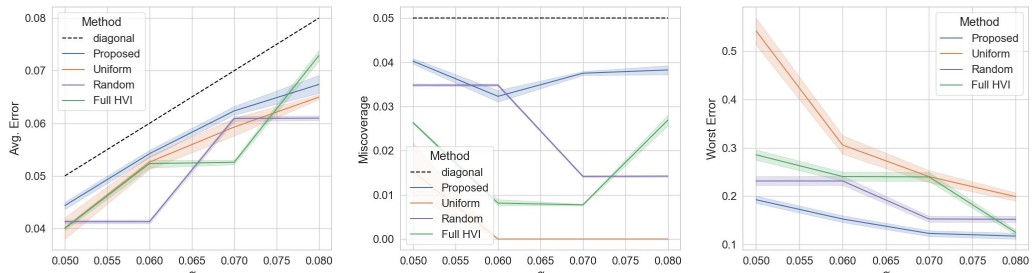

Figure C.2: Three-objectives, in the task of robustness with selective classification (50 random splits): average error limited is by $\alpha_1 \in [0.05, 0.08]$, miscoverage limited by $\alpha_2 = 0.05$, and worst accuracy minimized.

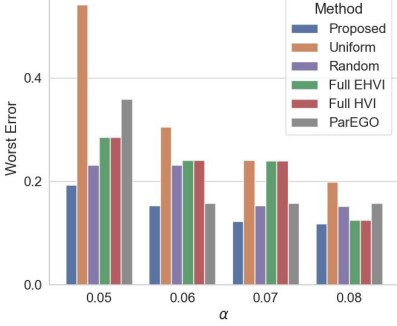

Figure C.3: Three objectives, additional baselines - presenting the scores obtained for the free objective - worst accuracy.

of the dense grid approach with $N = 50$. This indicates that using our proposed method we can significantly decrease the required budget without scarifying performance.

**Demonstration of BO Selection.** We show the outcomes of the proposed BO procedure across different tasks in Fig. C.5. The reference point defined in (10) is marked in green, and the boundaries of the region of interest are depicted by the dashed lines. The blue points correspond to the configurations in the initial pool $\mathcal{C}_0$, while the red points correspond to the configurations selected by our BO procedure. We see that the specified region is significantly smaller compared to the entire front. Moreover, we observe that through BO we obtain a dense set of configurations that is confined to the region of interest as desired.

**Influence of $\delta'$.** We examine the influence of $\delta'$, which determines the boundaries of the region of interest. Figure. C.6 shows the scores obtained for different values of $\delta'$. We observe that in most cases there is no noticeable difference in the performance with respect to $\delta'$, indicating that our method is generally insensitive to this choice.

**One sided bound.** We compare the proposed method to the case that the BO search is constrained by a one-sided bound at the upper limit defined in (7). Fig. C.7 shows the values of the free objective across tasks. We see that in most cases performing the search in the defined region of interest is preferable over a single-sided bound. This shows the benefit of removing low risk, inefficient configurations from the search space (the green section in Fig. E.2).

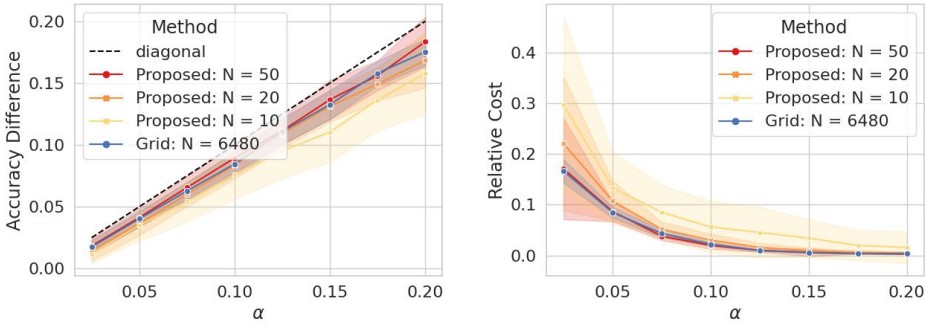

Figure C.4: Results of the proposed method over AG News for different number of evaluations, and with a grid of thresholds. Results are averaged over 50 random splits. Accuracy reduction is controlled and cost is minimized.

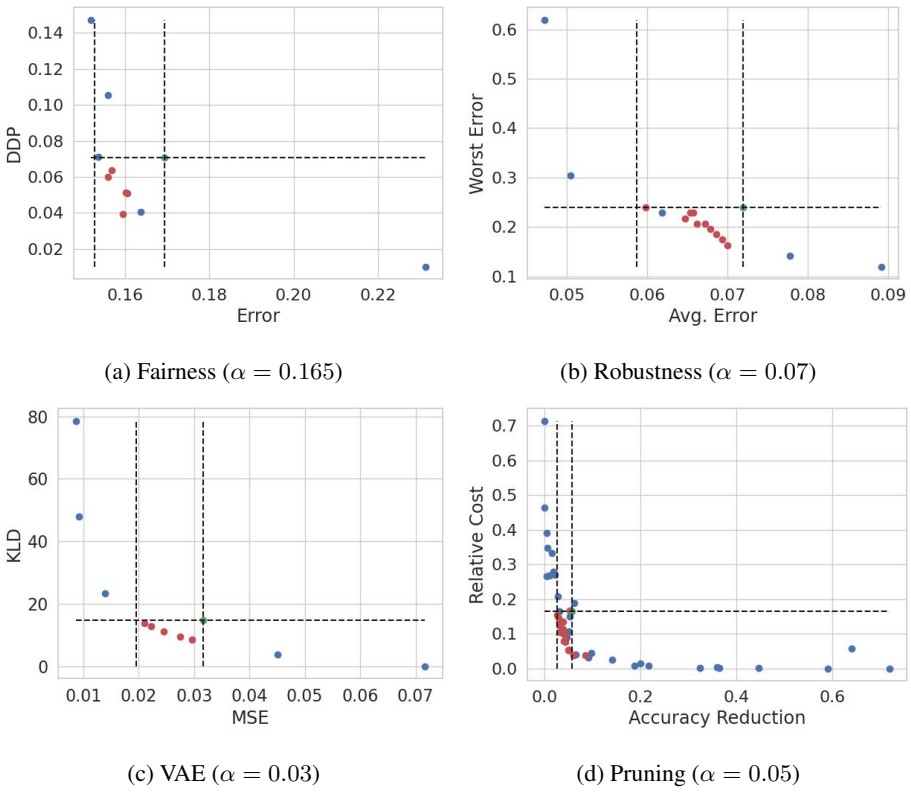

(a) Fairness ($\alpha = 0.165$)

(b) Robustness ($\alpha = 0.07$)

(c) VAE ($\alpha = 0.03$)

(d) Pruning ($\alpha = 0.05$)

Figure C.5: Demonstration of the selection outcomes of the proposed BO: the green point is the defined reference point, the blue points correspond to the initial set of configurations, and the red points correspond to selected configurations. Dashed lines enclose the region of interest.

## D   ALGORITHMS

Our proposed guided BO procedure and the overall method, are summarized in Algorithms D.1, and D.2, respectively.

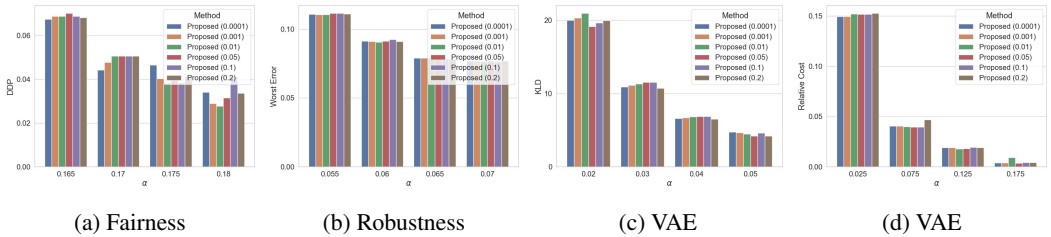

|   (a) Fairness   |   (b) Robustness   |   (c) VAE   |   (d) VAE   |

Figure C.6: Influence of $\delta'$. Showing the scores of the free objective for different values of $\delta'$, which controls the width of the region of interest, defined in (7)

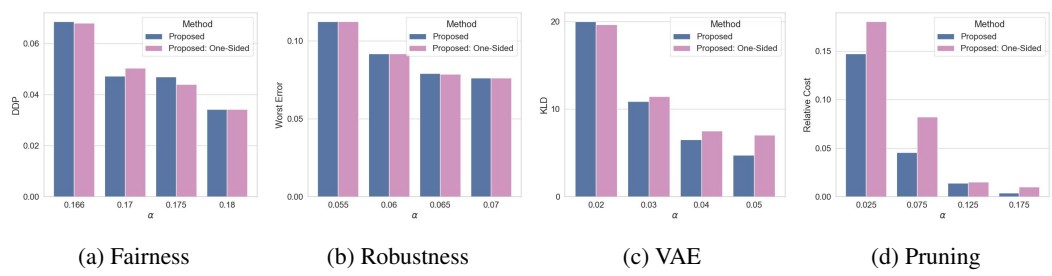

|   (a) Fairness   |   (b) Robustness   |   (c) VAE   |   (d) Pruning   |

Figure C.7: Ablation study - one sided bound instead of the defined two-sided region. Presenting the scores obtained for the free objective.

# E   MATHEMATICAL DETAILS

## E.1   DERIVATION OF THE REGION OF INTEREST

Suppose the loss is bounded above by 1, then Hoeffding's inequality (Hoeffding, 1994) is given by:

$$P\left(\hat{\ell}(\boldsymbol{\lambda}) - \ell(\boldsymbol{\lambda}) \leq -t\right) \leq e^{-2nt^2}. \tag{16}$$

and

$$P\left(\hat{\ell}(\boldsymbol{\lambda}) - \ell(\boldsymbol{\lambda}) \geq t\right) \leq e^{-2nt^2}. \tag{17}$$

for $t > 0$. Taking $u = e^{-2nt^2}$, we have $t = \sqrt{\frac{\log(1/u)}{2n}}$, hence:

$$P\left(\hat{\ell}(\boldsymbol{\lambda}) - \ell(\boldsymbol{\lambda}) \leq -\sqrt{\frac{\log\left(1/u\right)}{2n}}\right) \leq u. \tag{18}$$

and

$$P\left(\hat{\ell}(\boldsymbol{\lambda}) - \ell(\boldsymbol{\lambda}) \geq \sqrt{\frac{\log\left(1/u\right)}{2n}}\right) \leq u. \tag{19}$$

This implies an upper confidence bound

$$\ell_{\mathrm{HF}}^+(\boldsymbol{\lambda}) = \hat{\ell}(\boldsymbol{\lambda}) + \sqrt{\frac{\log\left(1/u\right)}{2n}} \tag{20}$$

and a lower confidence bound

$$\ell_{\mathrm{HF}}^-(\boldsymbol{\lambda}) = \hat{\ell}(\boldsymbol{\lambda}) - \sqrt{\frac{\log\left(1/u\right)}{2n}}. \tag{21}$$

---

**Algorithm D.1** Testing-Guided Bayesian Optimization

**Definitions:** $\mathcal{D}_{\text{val}} = \{X_i, Y_i\}_{i=1}^{k}$ is the validation data. $\ell_1, \ldots, \ell_c$ and $\ell_{\text{free}}$ are the objective functions, $g_1, \ldots, g_c$ and $g_{\text{free}}$ are their associated surrogate models. $\ell_1^{\text{low}}, \ldots, \ell_c^{\text{low}}$ and $\ell_1^{\text{high}}, \ldots, \ell_c^{\text{high}}$ are the lower and upper bounds, respectively, for the first $c$ objectives. $\mathcal{C}_0 = \{\boldsymbol{\lambda}_0, \ldots, \boldsymbol{\lambda}_{N_0}\}$ is an initial pool of configurations and $\mathcal{L}_0 = \{\hat{\boldsymbol{\ell}}(\boldsymbol{\lambda}_1), \ldots, \hat{\boldsymbol{\ell}}(\boldsymbol{\lambda}_{N_0})\}$ are the associated objectives. $N$ is our total budget.

1: **function** BO($\mathcal{D}_{\text{val}}, \boldsymbol{\ell}, \mathcal{C}_0, \mathcal{L}_0, \{\ell_1^{\text{low}}, \ldots, \ell_c^{\text{low}}\}, \{\ell_1^{\text{high}}, \ldots, \ell_j^{\text{high}}\}, N$)
2:      $N_{\max} \leftarrow N - N_0$
3:      $\mathbf{r} \leftarrow \left( \ell_1^{\text{high}}, \ldots, \ell_c^{\text{high}}, \max_{\boldsymbol{\lambda} \in \mathcal{C}_0} \ell_{\text{free}}(\boldsymbol{\lambda}) \right)$          ▷ Initialize reference point.
4:      **for** $n = 0, 1, 2, \ldots, N_{\max} - 1$ **do**
5:          Fit $\hat{\boldsymbol{g}}$ on $(\mathcal{C}_n, \mathcal{L}_n)$          ▷ Fit surrogate models.
6:          $r_{c+1} \leftarrow \min_{\boldsymbol{\lambda} \in R^{\text{low}}} \hat{g}_{\text{free}}(\boldsymbol{\lambda})$          ▷ Update reference point.
7:          $\hat{\mathcal{P}} \leftarrow \text{ParetoFront}(\mathcal{L}_n)$          ▷ Filter Pareto front.
8:          $\boldsymbol{\lambda}_{n+1} = \arg\max_{\boldsymbol{\lambda}} HVI(\hat{\boldsymbol{g}}(\boldsymbol{\lambda}), \hat{\mathcal{P}}; \mathbf{r})$.          ▷ Optimize acquisition function.
9:          Evaluate $\hat{\boldsymbol{\ell}}(\boldsymbol{\lambda}_{n+1})$ over $\mathcal{D}_{\text{val}}$.          ▷ Evaluate new configuration.
10:         $\mathcal{C}_{n+1} \leftarrow \mathcal{C}_n \cup \boldsymbol{\lambda}_{n+1}$.          ▷ Add new configuration.
11:         $\mathcal{L}_{n+1} \leftarrow \mathcal{L}_n \cup \hat{\boldsymbol{\ell}}(\boldsymbol{\lambda}_{n+1})$.          ▷ Add new objective values.
12:      $\mathcal{C}^{\text{BO}} \leftarrow \mathcal{C}_{N_{\max}}$
13:      **return** $\mathcal{C}^{\text{BO}}$

---

**Algorithm D.2** Configuration Selection

**Definitions:** $f$ is a configurable model set by an hyperparameter $\boldsymbol{\lambda}$. $\mathcal{D}_{\text{val}} = \{X_i, Y_i\}_{i=1}^{k}$ and $\mathcal{D}_{\text{cal}} = \{X_i, Y_i\}_{i=k+1}^{k+m}$ are two disjoint subsets of validation and calibration data, respectively. $\{\ell_1, \ldots, \ell_c\}$ are constrained objective functions, and $\ell_{\text{free}}$ is a free objective. $\{\alpha_1, \ldots, \alpha_c\}$ are user-specified bounds for the constrained objectives. $\Lambda$ is the configuration space. $\delta$ is the tolerance. $N$ is the optimization budget. PARETOOP-TIMALSET returns Pareto optimal points.

1: **function** SELECT($\mathcal{D}_{\text{val}}, \mathcal{D}_{\text{cal}}, \Lambda, \{\alpha_1, \ldots, \alpha_c\}, \delta, N$)
2:      Compute $\ell_i^{\text{low}}, \ell_i^{\text{high}}$ for $i \in \{1, \ldots, c\}$ based on (7) and (8)          ▷ Determine the region of interest.
3:      $\mathcal{C}_0, \mathcal{L}_0 \leftarrow$ Randomly sample an initial pool of configurations          ▷ Generate an initial pool.
4:      $\mathcal{C}^{\text{BO}} \leftarrow \text{BO}(\mathcal{D}_{\text{val}}, \boldsymbol{\ell}, \mathcal{C}_0, \mathcal{L}_o, \{\ell_1^{\text{low}}, \ldots, \ell_c^{\text{low}}\}, \{\ell_1^{\text{high}}, \ldots, \ell_j^{\text{high}}\}, N)$          ▷ BO via Algorithm D.1.
5:      $\mathcal{C}^{\text{p}} \leftarrow \text{PARETOOPTIMALSET}(\mathcal{C}^{\text{BO}})$          ▷ Filter Pareto points.
6:      Compute $p_{\boldsymbol{\lambda}}^{\text{val}}$ over $\mathcal{D}_{\text{val}}$ for all $\boldsymbol{\lambda} \in \mathcal{C}^{\text{p}}$          ▷ Compute approximated p-values.
7:      $\mathcal{C}^{\text{o}} \leftarrow$ Order configurations according to increasing $p_{\boldsymbol{\lambda}}^{\text{val}}$          ▷ Order configurations.
8:      Compute $p_{\boldsymbol{\lambda}}^{\text{cal}}$ over $\mathcal{D}_{\text{cal}}$ for all $\boldsymbol{\lambda} \in \mathcal{C}^{\text{o}}$          ▷ Compute p-values.
9:      Apply FST: $\mathcal{C}^{\text{valid}} = \{\boldsymbol{\lambda}^{(j)} : j < J\}, \ J = \min_j \{j : p_{\boldsymbol{\lambda}}^{\text{cal}} \geq \delta\}$          ▷ Apply FST.
10:      $\boldsymbol{\lambda}^* = \min_{\boldsymbol{\lambda} \in \mathcal{C}^{\text{valid}}} \ell_{\text{free}}(\boldsymbol{\lambda})$          ▷ Choose best-performing configuration.
11:      **return** $\boldsymbol{\lambda}^*$

---

In addition, we can use Hoeffding's inequality to derive a valid p-value under the null hypothesis $H_{\boldsymbol{\lambda}} : \ell(\boldsymbol{\lambda}) > \alpha$. By (18), we get:

$$P\left( \hat{\ell}(\boldsymbol{\lambda}) - \alpha \leq -\sqrt{\frac{\log(1/u)}{2n}} \right) \leq P\left( \hat{\ell}(\boldsymbol{\lambda}) - \ell(\boldsymbol{\lambda}) \leq -\sqrt{\frac{\log(1/u)}{2n}} \right) \leq u. \tag{22}$$

For $\hat{\ell}(\boldsymbol{\lambda}) < \alpha$, we rearrange (22) to obtain:

$$P\left( e^{-2n(\alpha - \hat{\ell}(\boldsymbol{\lambda}))^2} \leq u \right) \leq u, \tag{23}$$

which implies that $p_{\boldsymbol{\lambda}}^{\text{HF}} := e^{-2m\left(\alpha - \hat{\ell}(\boldsymbol{\lambda})\right)_+^2}$ is super-uniform, hence is a valid p-value. Comparing $p_{\boldsymbol{\lambda}}^{\text{HF}}$ to $\delta$, yields the maximum empirical loss $\hat{\ell}(\boldsymbol{\lambda})$, evaluated over a calibration set of size $m$, which can pass the test with significance level $\delta$:

$$\alpha^{\text{max}} = \alpha - \sqrt{\frac{\log(1/\delta)}{2m}}. \tag{24}$$

This can be equivalently obtained from the upper bound (20).

A tighter alternative to Hoeffding p-value was proposed in (Bates et al., 2021) based Hoeffding and Bentkus inequalities. The Hoeffding-Bentkus p-value is given by:

$$p_{\boldsymbol{\lambda}}^{\text{HB}} = \min\left(\exp\{-mh_1(\hat{\ell}(\boldsymbol{\lambda}) \wedge \alpha, \alpha)\}, e\mathbb{P}\left(\text{Binom}(m, \alpha) \leq \lceil m\hat{\ell}(\boldsymbol{\lambda})\rceil\right)\right) \tag{25}$$

where $h_1(a, b) = a\log(\frac{a}{b}) + (1-a)\log(\frac{1-a}{1-b})$. Note that for a given $\delta$ we can numerically extract from (25) the upper and lower bounds corresponding to a $1 - \delta$ confidence interval, and use it to define the region of interest as in (7).

### E.2 PROOF OF PROPOSITION 5.1

The proof is based on (Angelopoulos et al., 2021; Laufer-Goldshtein et al., 2023), which we repeat here for completeness.

*Proof.* Recall that $\mathcal{D}_{\text{val}}$ and $\mathcal{D}_{\text{cal}}$ are two disjoint, i.i.d. datasets. Therefore, $\mathcal{D}_{\text{cal}}$ is i.i.d. w.r.t the returned configuration set optimized in Algorithm D.1 over $\mathcal{D}_{\text{val}}$.

We now prove that the testing procedure returns a set of valid configurations with FWER bounded by $\delta$. Let $H_{\boldsymbol{\lambda}'}$ be the first true null hypothesis in the sequence. Given that $p_{\boldsymbol{\lambda}'}$ is a super uniform p-value under $H_{\boldsymbol{\lambda}'}$, the probability of making a false discovery at $\boldsymbol{\lambda}'$ is bounded by $\delta$. This means that the event that $H_{\boldsymbol{\lambda}'}$ is rejected (false discovery) occurs with probability lower than $\delta$. According to the sequential testing procedure, all other $H_{\boldsymbol{\lambda}}$ that follow are also rejected (regardless of if $H_{\boldsymbol{\lambda}}$ is true or not). Therefore the probability of making any false discovery is bounded by $\delta$, which satisfies the FWER control requirement.

$\square$

### E.3 HYPERVOLUME

An illustration of the hypervolume defined in (3) is given in Fig. E.1 for the 2-dimensional case. It can be seen that the hypervolume is equivalent to the volume of the union of the boxes created by the Pareto optimal points.

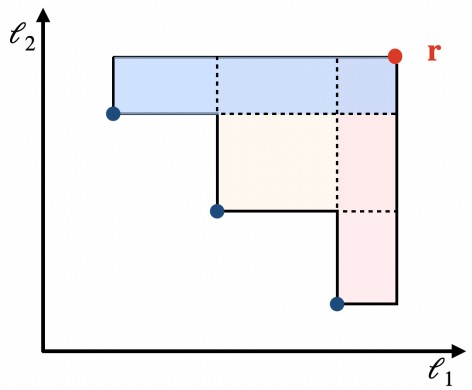

Figure E.1: An illustration of the hypervolume in the 2-dimensional case. The reference point is marked in red and three Pareto optimal points are marked in blue.

### E.4 ILLUSTRATION

Figure E.2 shows the partition of the Pareto front to the different regions, and demonstrates the difference between Pareto Testing and the proposed method.

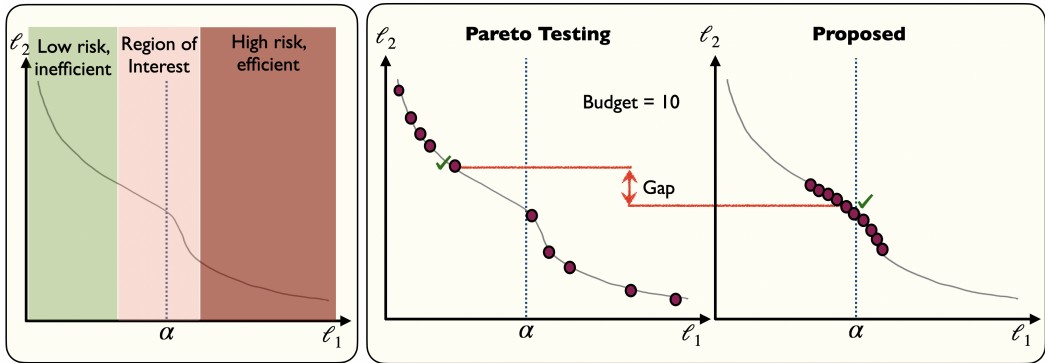

Figure E.2: Left: Illustration of the different parts of the Pareto front. The green region consists of configurations that are low risk ($\ell_1 \ll \alpha$) but inefficient in terms of the free objective $\ell_2$. The brown region consists of configurations that are efficient but high risk ($\ell_1 \gg \alpha$) and cannot pass the test. In the middle we define the region of interest containing configurations that are likely to be both valid and efficient. Right: comparing the proposed method to Pareto Testing for optimization budget $N = 10$. In Pareto Testing there is no control on the distribution of the configurations on the front, while our method focuses on the region of interest. As a result, there is a gap in the minimization of $\ell_2$ for the chosen valid configuration (marked by v) in favor of the proposed method.

