# OpenReview forum: "Risk-Controlling Model Selection via Guided Bayesian Optimization"
_ICLR.cc/2024/Conference — Submitted to ICLR 2024_

### Official Review · Reviewer_kvnt · 2023-10-23

**Soundness:** 3 good
**Presentation:** 3 good
**Contribution:** 3 good
**Rating:** 5
**Confidence:** 3

**Summary:**

This paper adds to the growing body of literature with respect to distribution free uncertainty quantification and risk control.  While most work in this area descends from the line of work concerned with Conformal Prediction, the field was really expanded and energized by the work of Angelopoulos et al (2021) called Learn Then Test, which introduces a method for performing hypothesis/model/parameter selection based on multi-hypothesis testing (MHT) and family-wise error rates.  One clear inefficiency of this LTT procedure is that statistical power is lost as more hypotheses are considered; this issue was directly addressed in the work of Laufer-Goldshtein et al (2022), wherein an initial step is added in order to narrow some larger configuration space down to a set of hypotheses that are worth considering during MHT.

The current submission aims to further build on the work of Laufer-Goldshtein.  Their algorithm modifies Laufer-Goldshtein in two key ways, by: 1) further narrowing the hypothesis search space by defining a region of interest based on various loss calculations on a validation set prior to testing 2) using Bayesian Optimization to efficiently identify configurations that both are Pareto-optimal and lie within the region of interest.  They apply their algorithm to important problems like fair and robust classification, image generation, and large model pruning.

**Strengths:**

The authors address an important problem with this work.  Though risk control techniques are a key ingredient in responsible machine learning deployment, they are limited by the fact that they either have to ignore a large part of the search space or sacrifice statistical power in order to consider all reasonable hyperparameter settings.  While this was addressed in the previous Pareto Testing paper, more work in this direction is great.  The idea of further reducing the search space using the validation set and BO seems quite natural.

**Weaknesses:**

-I find the definition of the region of interest to be under-motivated.  It is not clear why the region of interest is defined in this way; it would be useful to further motivate the definition by some worked example or analysis, and the definition could be compared/contrasted with other possible ways for defining the region.  It may be the case that the algorithm is primarily focused on the case where objectives are conflicting; if this is so, it should be stated more clearly, along with a description of how the algorithm will perform when this is not the case.

-While the authors seem to be motivated by the problem of “expansive high-volume configuration spaces”, their experimental applications seem to have configuration spaces that are of a similar size as those in previous work.  While performing better in these spaces is useful, it would also be interesting to see that this algorithm enables some new class of problem(s) to be addressed via MHT.

-In addition, the experiments section does not make a strong case for the current algorithm.  There are several reasons for this:

1) Unexplained hyperparameter choices - how is $\delta'$ chosen?  This seems very important, and I see no mention of this in the experiments section (also see questions below).

2) Unclear how limits of x-axes were chosen - Why should we only be concerned with $\alpha \in [0.165, 0.18]$ for Figure 3(a) or $\alpha \in [0.055, 0.07]$ in Figure 3(b)?  These choices seem arbitrary, and make it hard to takeaway that the method is generally better than the simple baseline.

3) Insignificant results based on standard deviation - Figure 3(a) and 3(b) do not show the proposed algorithm to be much better than a fairly naive baseline in what seems to be a highly-controlled setting (see (1) and (2) above).

-Further building on the previous comment, I think the experiments section would benefit from a focus on less results that are more clearly explained.  Even examining the details in the appendix leaves many experimental choices a mystery.

-Since it is not clear why this definition of the region of interest is of particular significance (either via analysis or empirical results), I find the contribution to be incremental and short of the acceptance threshold, although I could be convinced otherwise by other reviews and/or author rebuttals (see questions below).

**Questions:**

Below are the major questions that I am left with after reviewing the paper.  Since the authors are solving an important problem, I could be convinced to raise my score if I can come to a better understanding of why this is a particularly well-motivated and effective solution:

-Can you give a more detailed explanation of the rationale behind the definition of the ROI?  When is it expected to be most effective, and when may it fail?

-Can you more concretely characterize the efficiency of your method in relation to Pareto Testing?  How much compute or time is actually being saved, and how much performance is lost compared to full Pareto Testing?  I think this should be covered more thoroughly in the experiments section, as this seems to be the main comparison made prior to the experiments section.

-How is $\delta'$ chosen? This seems like an important parameter, but I cannot find a reference to its value in the experiments, nevermind how that value is chosen.  Right now, it seems possible that equation (7) is not actually that important, but instead the ROI is just based on a heuristic choice of tolerance around $\alpha^{max}$.  An ablation with respect to this parameter would also be helpful.

---

> ### Author Response · Authors · 2023-11-20
> **Response to reviewer kvnt (1/2)**
>
> Thank you for your helpful and constructive review.
>
> > On the rational behind the definition of the ROI and expected limitations
>
> We assume that there is a limited budget for searching the configuration space since each evaluation is very expensive (requires retraining of the model). We want to focus the optimization efforts on finding configurations that are: (i) likely to pass the test (the risk is not overly high), and are (ii) effective with respect to the free objective. The defined $\alpha^\textrm{max}$ specifies the limiting value that, if observed during testing, leads to rejecting the null hypothesis and declaring that the configuration is valid. Since the objectives are conflicting, as we get close to this limiting value, we improve in terms of the free objective. Ideally, we would like to find the configurations whose expected value is exactly $\alpha^\textrm{max}$. We include in the region of interest all the configurations that are likely to be associated with this ideal configuration. Specifically, we exclude configurations if the probability to correspond to an expected value $\alpha^\textrm{max}$ is lower than $\delta’$. These configurations are not relevant since they are likely to fail the test or they do not best minimize the free objective.  To improve clarity, we added clarifications around Eqs. (5), (6) and (7) and an illustration demonstrating these principles in Fig. E2.
>
> Regarding the effectiveness, it depends on the given budget and the shape of the frontier. In the limit of a large budget, sampling the entire front or only a portion of it, will be comparable. In addition, if the derivative of the free objective with respect to the constrained objective is small, the improvement in the free objective as we get closer to the limit is small as well. However, even in these cases the proposed method is not supposed to fail or perform worse than the other baselines.
>
> > On non-conflicting objectives
>
> Indeed, if the objectives are not conflicting the defined region of interest may not be relevant. This case can be trivially solved as a constrained optimization problem, and the solution can be statistically tested and verified without any concern that it will not pass the test. Only when the objectives are conflicting, the minimizing solution satisfies the constraint tightly, and we need to find a set of configurations around this point in order to increase the chances that we would be able to find a valid configuration.
>
> > On “expansive high-volume configuration spaces” not demonstrated
>
> We corrected this to “expansive configuration spaces”.  We indeed present examples with configuration space similar in size to what was previously shown. The major difference is that we demonstrate expensive configuration spaces that require retraining of large models per each new configuration. In previous works, the configurations were set by hyperparameters that affect the way the model is used after the model is already trained, which is significantly less expensive, and can be performed with a relatively large budget.
>
> > On the choice of $\delta’$
>
> We added explanations about the meaning and the choice of $\delta’$ in the revised manuscript. This hyperparameter controls the size of the region around $\alpha^\textrm{max}$.  We exclude configurations that correspond to $\alpha^\textrm{max}$ with probability that is less than $\delta’$. This is an empirical choice with a tradeoff between enlarging the region of interest to include other potential configurations and increasing the density of the configurations that are found close to $\alpha^\textrm{max}$.
>
> We use $\delta’=0.0001$.  Following your comment, we added an experiment to examine the influence of different $\delta’$ values (see Fig. C.5.). We observe that most of the time the exact choice of $\delta’$ does not lead to significant differences in the performance. Note that the width of the region is also influenced by the size of the validation data. When $k$ increases, the width decreases as we have more confidence in the observed empirical losses of being reflective of the actual expected loss.

---

> ### Author Response · Authors · 2023-11-20
> **Response to reviewer kvnt (2/2)**
>
> > On the chosen ranges of $\alpha$ values
>
> We added clarifications on the choice of these values in the revised manuscript. We define the range according to the values observed for the initial pool of configurations that is generated at the beginning of the BO procedure. The minimum and the maximum edge points obtained for each task appear in table B.1 and are also observed in the examples shown on Fig. C.4.  We choose values in between these extreme edge points, but not too close to either side, since too small values may not be statistically achievable and too large values are trivially satisfied (with tighter control not significantly improving the free objective). Note that in the chosen range even small differences in the limit values along the constrained axis, result in a significant difference for the free objective. As a general note, this choice is task and model specific, and for other tasks (for example pruning) the range is much wider.
>
> > On insignificant results for fairness and robustness
>
> We show in the experimental results that the proposed method is consistently better compared to all other baselines, almost everywhere across tasks and $\alpha$ values. It is true that for certain tasks the naïve baselines perform well but these baselines perform much worse in other settings. The naïve baselines spread the configurations uniformly or randomly in the configuration space. The multi-objective baselines spread them all over the Pareto front. It can happen that in some cases the resulting set contains configurations that reside in proximity to the testing limit, hence leading to efficient testing outcomes. However, more often it happens that the configurations are far from the limit, resulting in reduced efficiency. The proposed method always finds a dense set of configurations around the desired limit, thus is much more stable and efficient across different settings. We added an illustration in Fig. E2. to demonstrate this principle.
>
> > On missing experimental details
>
> Following your comment, we edited the experimental section and added further details and clarifications.
>
> > On the efficiency of the proposed method with respect to Pareto Testing
>
> We can compare the efficiency of the proposed method with respect to Pareto Testing from two viewpoints. The first viewpoint compares testing  efficiency (quantified via a  better minimization of the free objective) for a  given optimization budget. Pareto testing suggests first finding a sampled version of the Pareto front, and then orders and tests these configurations via sequential testing. We claim that most of the Pareto front is irrelevant, and we should focus only on the part of the front that is near the testing limit. For a given finite budget, our method yields a denser set of configurations near the limit, and therefore is expected to obtain tighter control. This is demonstrated in Fig. 3, 4, C1 and C2 and is illustrated in Fig. E.2.
> The second viewpoint is the budget required to match the ideal performance of Pareto Testing with a dense grid of configurations. As shown in Fig. C.3. with a budget of 50 configurations we can match the performance of Pareto testing with a dense grid of over 6K configurations.  From this viewpoint, our method has a computational advantage and can be applied to tuning hyperparameters that affect the training of large models, which was not exemplified before in Pareto Testing.

---

> > ### Comment · Reviewer_kvnt · 2023-11-21
> >
> > Thank you for your replies and the additional experiments and updated manuscript.  The clarifications and experiments around $\delta'$ are especially helpful.
> >
> > My concerns remain regarding the empirical results.  The results do not seem to be significantly better than the very naive uniform baseline.  I wonder if this baseline would get much stronger if the user narrowed down the uniform hyperparameter space a bit, which seems reasonable (e.g. if it was uniform on [0.2, 0.8], instead of [0.0, 1.0] for fairness).
> >
> > Also, I am curious how Figure 3 would look including the random baseline (which could also likely be improved with little effort by choosing a sensible hyperparameter space).  The random baseline seems to be at least as good as (and probably better than) the uniform baseline based on C.1, so it's not clear why only the uniform baseline made it to Figure 3.

---

> > > ### Author Response · Authors · 2023-11-21
> > > **Response**
> > >
> > > Thank you for reviewing our response and the revised paper.
> > >
> > > We added additional figures in the Supplementary Material where the free objective scores are presented for all methods in all 5 examined settings. It can be seen that both Uniform and Random perform poorly for the following tasks: VAE, pruning and selective classification and robustness.
> > >
> > > Narrowing down the search space might help in certain cases. However, it is unclear how to determine the exact bounds that are reasonable for each hyperparameter in each task.  We do not know in advance how the variation in the hyperparameter value affects the objective functions. If we choose the lower limit too high, we might not be able to find any valid configuration, especially for small $\alpha$. For example, for selective classification with a miscoverage constraint of 5%, we cannot know what is a reasonable starting value (0.5, 0.7, 0.8 or else) for which we would be sure to find a valid configuration that satisfies the constraint. In addition, when the dimension of the hyperparameter combination is bigger than one, narrowing the search space of all hyperparameters is not necessarily reasonable.  For example, in the pruning task we have three hyperparameters that control three pruning dimensions. It might be that the optimal configuration is pruning in two dimensions while leaving out the third dimension, which is excluded from the search space if we narrow down the ranges of all three dimensions.

---

> > > > ### Comment · Reviewer_kvnt · 2023-11-22
> > > >
> > > > Ok, thank you for the response.  I am still unsure why only the uniform baseline is presented in Figure 3, since by my count the random baseline outperforms it on 13 of 16 $\alpha$ measurements in Figure C.1.

---

> > > > > ### Author Response · Authors · 2023-11-22
> > > > > **Thank you for your feedback**
> > > > >
> > > > > Thank you for pointing this out. We updated Fig. 3 to include the Random baseline.

---

### Official Review · Reviewer_v3KF · 2023-10-30

**Soundness:** 2 fair
**Presentation:** 3 good
**Contribution:** 3 good
**Rating:** 6
**Confidence:** 2

**Summary:**

The authors incorporate risk control into multiobjective Bayesian optimization (MOBO). The problem setting assumes $c$ objective functions with constraints that are desired to hold with high probability, and $1$ unconstrained objective function that is desired to be minimized. The proposed method is as follows: 1) Define a fixed region of interest based on the desired significance level and sizes of validation and calibration datasets; 2) run a MOBO algorithm with reference point chosen each iteration based on the region of interest and posterior mean of the unconstrained objective function; 3) do hypothesis testing to pick the final configuration so that the constraints hold with high probability. The proposed method is empirically evaluated.

**Strengths:**

1. Incorporating risk control to do constrained MOBO is novel and likely to be of interest to the BO community.
2. The empirical settings are realistic and interesting, including settings on fairness and robustness, and VAEs and transformers.
3. The paper is generally written well.

**Weaknesses:**

1. **Empirical investigation does not properly evaluate main contribution.** As far as I can tell, the main novel contribution is the method of defining the region of interest and the reference point, after which an existing MOBO algorithm is used, and an existing algorithm for testing the validity of the returned configuration is used. The method of defining the region of interest and the reference point is not theoretically backed, which means that its usefulness is supported purely by the empirical results. However, the empirical investigation compares the entire procedure to existing MOBO algorithms (under only the Pruning setting), and the existing MOBO algorithms do not undergo the testing procedure. It is unclear how much of the performance gain is due to the main contribution (region of interest and reference point) and how much is due to the testing. The paper should include ablation studies to empirically support the claim that the main contribution is useful. How does the proposed method perform without defining the region of interest? How about if it is only a lower bound instead of an interval? It may turn out that most of the performance gain is due to the testing, rendering the main contribution not useful.

    Furthermore, it is not clear why no previous baselines are tested in the results for Figures 3 and 5. The empirical evaluation is the only support for the proposed method, and it needs to be more comprehensive.

2. **Clarity issues**. See Questions section.

**Questions:**

1. In the second paragraph of Sec. 3 Problem Formulation, should it be $L_i : \mathcal Y \times \mathcal Y \times \Lambda \rightarrow \mathbb R$ instead? In this case, $L_i$ in the preceding expectation should be $L_i(f_\lambda(X), Y, \lambda)$.

2. In Equation (7), should the formula for $\ell_{\text{low}}$ and $\ell_{\text{high}}$ be swapped?

3. In Algorithm D.1 line 9, how exactly is $\mathbf \ell (\lambda_{n+1})$ evaluated? Using $\mathcal D_{\text{train}}$ or $\mathcal D_{\text{cal}}$ ? From the line "while running the BO procedure we only have access to finite-size validation data $|\mathcal D_{\text{val}}| = k$", it sounds like it should be $\mathcal D_{\text{cal}}$. But $\mathcal D_{\text{cal}}$ is not passed into Algorithm D.1, hence the confusion.

4. Why aren't the previous MOBO algorithms tested in the results in Figures 3 and 5? What about the $n=1$ case prevents these baselines from being tested?

---

> ### Author Response · Authors · 2023-11-20
> **Response to reviewer v3KF**
>
> Thank you for your helpful and constructive review.
>
> > On comparison to baselines that perform multi-objective-optimization without testing
>
> Thank you for pointing this out. We understand we did not explain clearly the exact baselines that were compared. All baselines consist of two stages: optimization followed by testing. We do not include baselines that do not contain testing, since they were shown in [1] to lack valid risk control (they do not satisfy Eq. (1)). It was also shown in [1] that Pareto Testing is preferable over other testing mechanisms. In this paper, we compare our method to baselines that are all equivalent to Pareto Testing, while utilizing different methods for optimization in the first stage (and then perform testing in the second stage): uniform, random and different multi-objective optimization algorithms. Therefore, the testing mechanism is the same for all presented methods, and the improvement achieved by our method is related to the different optimization mechanism that is designed for improving testing efficiency.
>
> [1] Laufer-Goldshtein, Bracha, et al. "Efficiently Controlling Multiple Risks with Pareto Testing." The Eleventh International Conference on Learning Representations. 2022.‏
>
>  > On the performance without defining the region of interest
>
> The baselines of the multi-objective optimizers correspond to the case that no region is defined, and the entire Pareto front is recovered.
>
> > On the performance with a single limit
>
> Following your comment, we added an ablation study (see Fig. C.7.) to examine the case of a single limit instead of the defined region. This  is shown to be inferior with respect to the full region in most cases, as expected.
>
> > On notation in Section 3.
>
> We changed the notation following your suggestion.
>
> > On swapped limits in Eq. (7)
>
> We corrected the typo.
>
> > On the evaluation of the loss in Algorithm D.1.
>
> We use the validation data during BO. Following your comment, we added it as an input to the algorithm, and clarified its use for evaluating $\hat{\ell}$. The calibration data is used afterwards in the testing procedure.
>
> > On the comparison to MOBO baselines in Figs. 5 and 6.
>
> Following the reviewers’ comments, we added these comparisons. See extended results in Figs.  3, 4, C1 and C2.

---

> > ### Comment · Reviewer_v3KF · 2023-11-21
> >
> > Thanks for your response. Are the results for EHVI and ParEGO missing from Fig. 3?

---

> > > ### Author Response · Authors · 2023-11-21
> > > **Response**
> > >
> > > Thanks for your response. For the clarity of the figures, we put two baselines (uniform and HVI) in the line plots on Fig.3, and all baselines in the bar plots on Fig. C.1 (in the Appendix).

---

> > > ### Author Response · Authors · 2023-11-21
> > > **Response Cont.**
> > >
> > > Thank you again for reviewing our response and the revised paper.
> > >
> > > For your convenience, we added additional line plot figures in the Supplementary Material where the free objective scores are presented for all methods in all 5 examined settings.

---

> > > > ### Comment · Reviewer_v3KF · 2023-11-22
> > > >
> > > > Thanks for addressing my concerns. I have increased my score.

---

### Official Review · Reviewer_tf6K · 2023-10-31

**Soundness:** 2 fair
**Presentation:** 3 good
**Contribution:** 2 fair
**Rating:** 5
**Confidence:** 3

**Summary:**

The paper tackles the problem of finding a configuration adhering to user-specified limits on certain aspects whilst being useful with respect to various conflicting metrics. The main idea is to formulate as a multi-objective optimization problem with multiple hypothesis tests. The paper first proposes to identify a region of interest that limits the search space for candidate configurations to obtain efficient testing with less computation. The paper then presents a new BO process that can identify configurations that are Pareto optimal and lie in the region of interest. The paper presents experimental results on various problems including classification fairness, classification robustness, VAE, and transformer pruning.

**Strengths:**

+ The paper tackles a quite interesting and general problem. I think this generic problem could be very useful in different settings like fairness, robustness, etc.
+ The paper’s writing is generally clear and easy to understand when describing about the problem settings, the motivation, the related work and the experimental evaluation.
+ I think the key idea of the proposed algorithm seems to be sound and reasonable – however, there are some detailed information I found unclear (which I ask in the below section).
+ Different categories of problems are used in the experimental evaluation – this helps to understand the applications of the problem setting in this paper.

**Weaknesses:**

There are various unclear descriptions of the proposed technique. I list in the below some points that I found unclear:
+ The formulation of the region of interest seems unclear to me. I don’t understand how Eq. (5) – which is to compute the p-value - is derived. And is this inequality be able to apply for any loss function $l(\lambda)$? Are there any particular requirements regarding this loss function for Eq. (5) to hold? Eq. (6) is also suddenly introduced without clear explanation why $\alpha^\max$ is defined that way.
+ In Eq. (7), there is also no clear explanation on how the confidence interval is constructed that way. What is the role of $\delta’$ here? How can it be chosen in practice? What significant level is associated with this confidence interval?
+ I’m not sure if I missed it but it seems like there is no proof of Theorem 5.1?

One of the main weaknesses to me is actually in the experimental evaluation. Below are my concerns regarding the experimental evaluation:
+ Among 4 test problems, three problems (classification fairness, classification robustness, VAE) actually only have a single-dimensional hyperparameter.
+ For the baselines of problems of one single-dimensional hyperparameter, I don’t understand why Uniform is chosen as a baseline but not Random. Why don’t we compare with both Uniform and Random for all problems? And why we don’t compare with a standard multi objective optimization method that does not make use of the region of interest?
+ Finally, the range of the value $\alpha$ seems to be quite small to me, e.g., for the classification fairness problem, the range of $\alpha$ is only from 0.165 to 0.180 or for the robustness problem, the range of $\alpha$ is only from 0.055 to 0.070.

**Questions:**

Please find my questions in the Weakness section.

---

> ### Author Response · Authors · 2023-11-20
> **Response to reviewer tf6K**
>
> Thank you for your helpful and constructive review.
>
> > On the clarity of Eqs. (5), (6) and (7)
>
> Following your comment, we added clarifications around Eqs. (5), (6) and (7), and additional derivations and mathematical details in Appendix E. In short, for testing we receive an empirical loss evaluated over the calibration data, and we need to determine whether the configuration is valid or not.  For bounded losses, Hoeffding’s inequality quantifies the probability to obtain an empirical loss value $\hat{\ell}$  under a given expected loss $\ell$.  In appendix E, we show how to derive Eq. (5) from Hoeffding’s inequality. Eq. (6)  quantifies the value of $\hat{\ell}_1$ for which the p-value in (5) equals $\delta$. This is denoted as $\alpha^\textrm{max}$. In order to be valid the loss must be lower than $\alpha^\textrm{max}$. However, for effectively minimizing the free objective function, we should be  as close as possible to $\alpha^\textrm{max}$  (since increasing $\ell_1$ implies decreasing $\ell_2$). Therefore, we define in (7) the region of all configurations that are likely to be obtained under an expected value  $\alpha^\textrm{max}$. Specifically, we exclude configurations if their probability to be associated with $\alpha^\textrm{max}$ is lower than $\delta’$. The hyperparameter  $\delta’$ controls the region's width and is unrelated to the testing confidence level $\delta$. It is an empirical choice that trade-off between including more potential values, and obtaining a set of configurations that is dense around $\alpha^\textrm{max}$.  We added an experiment (see Fig. C.5.) showing that the method is generally  insensitive to the choice of $\delta’$.
>
> > On the lack of proof for Theorem 5.1.
>
> We added the proof on Appendix E.
>
> > On 3 out of 4 tasks with a single-dimensional hyperparameter
>
> Our experiments include two settings with a multi-dimensional hyperparameter. Specifically, we including the following five tasks:
>
> Classification fairness – 1 dimension
>
> Classification robustness – 1 dimension
>
> Selective classification and robustness – 2 dimensions
>
> VAE - 1 dimension
>
> Transformer pruning - 3 dimensions
>
> Note that compared to previous work (Pareto Testing),  we present four cases requiring model retraining for each new hyperparameter setting. This makes the optimization much more cost intensive and is enabled by our proposed approach, which focuses on a limited part of the Pareto front.
>
> > On missing baselines for $n=1$
>
> It is important to note that in the single dimension case, varying the hyperparameter value and retraining the model, results in configurations that reside close to the Pareto front. Therefore running multi-objective optimizers for these cases is not expected to be significantly different from the Uniform sampling baseline (though the configuration distribution may change). The Random baseline for $n>1$ is based on Latin Hypercube sampling, which is a semi-random sampling method that generates well-spread points in the configuration space. This is very close in spirit to the Uniform grid defined for $n=1$.
> Nevertheless, following your comment,  we added these baselines, as can be seen in Figs.  3, 4, C1 and C2.
>
> > On the chosen ranges of $\alpha$ values
>
> We added clarifications on the choice of these values in the revised manuscript. We define the range according to the values observed for the initial pool of configurations that is generated at the beginning of the BO procedure. The minimum and the maximum edge points obtained for each task appear in table B.1 and are also observed in the examples shown on Fig. C.4.  We choose values in between these extreme edge points, but not too close to either side, since too small values may not be statistically achievable and too large values are trivially satisfied (with tighter control not significantly improving the free objective). Note that in the chosen range even small differences in the limit values along the constrained axis, result in a significant difference w.r.t the free objective. As a general note, this choice is data and model specific, and for other tasks (for example pruning) the range is much wider.

---

> > ### Comment · Reviewer_tf6K · 2023-11-22
> >
> > Thank you for your response.
> >
> > Some of my concerns are addressed but some still remain.
> >
> > Firstly, I don't see the Random baseline added to Figure 3? Secondly, I still think the problems-under-test are still quite simple.
> >
> > Finally, regarding the assumptions used in deriving the proposed method, can you summarize clearly what assumptions you need to use? In the comments, the authors mention that a valid loss needs to have value less than \alpha^{\max}, but I look at the newly-added texts and the proof of Theorem 1, and it says the loss function needs to be less than 1. Furthermore, I'm just wondering if there are any requirements for the loss function to have some characteristics like Lipschitz continuous, or differentiable, or similarly?

---

> > > ### Author Response · Authors · 2023-11-22
> > > **Thank you for your feedback**
> > >
> > > Thank you for reviewing our response and the revised paper.
> > >
> > >
> > > > Missing Random baseline
> > >
> > > We updated Fig. 3 to include the Random baseline. For the sake of clarity, the other two baselines appear in Fig C.1.
> > >
> > >
> > > > I still think the problems-under-test are still quite simple.
> > >
> > > We believe that the problems-under-test represent important areas that are extensively explored by the ML community and can benefit from our method for reliable model selection under multiple risk constraints. These include: algorithmic fairness [1], robustness to spurious correlations [2], selective classification [3], rate and distortion in VAEs [4], and accuracy-cost trade-offs for pruning large-scale Transformer models [5]. It is also important to emphasize that this is the first time that risk control is demonstrated for tuning hyperparameters affecting the training process of large models. This significantly extends previous work in the field of uncertainty quantification, which focused on obtaining risk control by selecting threshold values over the outcomes of fixed trained models only.
> > >
> > >
> > > > On the assumptions regarding the loss function.
> > >
> > > The objective function (**the expected loss**) needs to be bounded by 1 in order to use the p-value defined in Eq. (5) or (25). Another p-value can be used in the unbounded case, as suggested in [6]. No other assumptions are required.
> > >
> > > During the testing procedure, we compute the p-value based on the empirical loss computed over the calibration set. If the resulting p-value is bellow $\delta$ we reject the null hypothesis, and declare that $\boldsymbol{\lambda}$ is valid. The value $\alpha^\textrm{max}$, defined in Eq. (6), is the **empirical loss** value for which the p-value exactly matches  $\delta$. This means that only if the empirical loss does not exceed $\alpha^\textrm{max}$ then $\boldsymbol{\lambda}$ is found to be risk-controlling (the **true expected loss** does not exceed $\alpha$ w.p. that is greater or equal to $1-\delta$, as specified in Eq. (1)).
> > >
> > > [1] Padh, Kirtan, et al. "Addressing fairness in classification with a model-agnostic multi-objective algorithm." Uncertainty in artificial intelligence. PMLR, 2021.‏
> > >
> > > [2] Izmailov, Pavel, et al. "On feature learning in the presence of spurious correlations." Advances in Neural Information Processing Systems 35 (2022): 38516-38532.
> > >
> > > ‏[3] Geifman, Yonatan, and Ran El-Yaniv. "Selective classification for deep neural networks." Advances in neural information processing systems 30 (2017).
> > >
> > > ‏[4] Bae, Juhan, et al. "Multi-Rate VAE: Train Once, Get the Full Rate-Distortion Curve." The Eleventh International Conference on Learning Representations. 2022.
> > >
> > > [5] Schuster, Tal, et al. "Consistent Accelerated Inference via Confident Adaptive Transformers." Proceedings of the 2021 Conference on Empirical Methods in Natural Language Processing. 2021.
> > >
> > > ‏[6] Angelopoulos, Anastasios N., et al. "Learn then test: Calibrating predictive algorithms to achieve risk control." arXiv preprint arXiv:2110.01052 (2021).‏

---

### Official Review · Reviewer_riWN · 2023-10-31

**Soundness:** 3 good
**Presentation:** 3 good
**Contribution:** 2 fair
**Rating:** 3
**Confidence:** 3

**Summary:**

The paper presents a new methodology for selecting model configurations/hyper-parameters based on user-defined risk constraints and performance metrics. By combining Bayesian Optimization (BO) with rigorous statistical testing, the authors aim to achieve efficient model selection. They introduce the concept of a "region of interest" in the objective space to optimize model parameters more efficiently. The proposed method has been shown to be versatile across various domains, including algorithmic fairness, robustness, and model efficiency.

**Strengths:**

The problem is of practical importance. In industry, the optimization cannot be done without any constraints and the "Risk-Controlling" factor in this paper is the primary focus. This paper proposed a relatively rigorous way to handle the problem with good experiment results. The work is further extended to the MOO setting which enables more potential applications of the proposed method.

**Weaknesses:**

1. The primary concern is whether the paper fits ICLR in general. The paper is focusing on statistical test based theories and proposing algorithms which are not directly related to representation learning.

2. This might be subjective, but many of the contents until page 5 are fairly standard, while later many details have to be put in the appendix. The narratives in section 6 and 7 are also not clear or confusing. The writing can be improved.

3. The sign of (7) should be changed

4. The practical use of the proposed method. The significance level \sigma is usually not the crucial factor, but rather performance or business metrics related to the objective function.

5. Though in the intro part the potentials of the proposed method are claimed to cover different areas, in the experiment section the test is only done on limited tasks.

**Questions:**

Please see the weakness part. Plus the following

1. How does the proposed method perform in situations with extremely high-dimensional hyperparameter spaces?

2. How sensitive is the proposed method to the initial definition of the "region of interest"? Is there a risk of introducing bias based on this region?

3. Given that Bayesian Optimization inherently deals with a trade-off between exploration and exploitation, how does the new approach ensure a balance, especially with the introduced "region of interest"?

4. In practical terms, what would be the computational overhead of the proposed method compared to traditional BO or other hyper-parameter optimization techniques?

---

> ### Author Response · Authors · 2023-11-20
> **Response to reviewer riWN**
>
> Thank you for your helpful and constructive review.
>
> > On mismatch to ICLR
>
> The paper belongs to the field of reliability of machine learning models, and how we can tune the model hyperparameters while satisfying multiple risk constraints. This is related to the broader statistical field of conformal prediction and risk control , which is the topic of many papers published in recent years in ICLR and other top ML conferences. In addition, we combine the fields of hyperparameter tuning, multi-objective optimization and Bayesian optimization, which are core techniques in building machine learning models. Our method is demonstrated on different applications involving classification and generative models on benchmark machine learning tasks.
>
> > On writing clarity
>
> We made major changes to the paper’s writing and clarified points that were unclear to the reviewers.
>
> > On the sign of (7)
>
> We corrected the typo.
>
> > On the practical use of the proposed method
>
> We agree that  performance or business metrics related to the objective function are crucial in many practical applications.  However, in many cases there are tradeoffs between these metrics (accuracy vs cost, fairness, robustness etc.), and it is important to constrain certain metrics while optimizing others. Our method presents a simple-yet-effective approach to solve this by a new guided BO method combined with statistical testing.  We guarantee $(\alpha,\delta)$-risk control, where $\alpha$ is the bound and $\delta$ is the significance level, reflecting the certainty level with respect to the randomness of the calibration data. If $\delta =0.1$, then we can guarantee that 90% of the time (over different calibration datasets) the risk does not exceed $\alpha$.
>
> > On experiments testing limited tasks compared to introduction
>
> We demonstrate the performance in all the tasks mentioned in the introduction: algorithmic fairness, robustness to spurious correlations, generation capabilities in VAEs and cost reduction in Transformer pruning.
>
> > On the performance in high-dimensional hyperparameter spaces.
>
> The proposed method consists of two stages: (i) Bayesian optimization, and  (ii) multiple hypothesis testing. The first stage is affected by the dimensionality of the hyperparameter space. We adopt Bayesian optimization techniques, which are effective in optimizing large hyperparameter spaces. For extremely high-dimensional hyperparameter spaces we can incorporate more advanced methods for effective hyperparameter optimization [1]. The second stage of statistical testing is performed over a proportion of the Pareto frontier in the objective space, and, therefore, is not affected by the hyperparameter dimension.
>
> [1] Daulton, Samuel, et al. "Multi-objective Bayesian optimization over high-dimensional search spaces." Uncertainty in Artificial Intelligence. PMLR, 2022.‏
>
> > On the risk of bias and sensitivity w.r.t. the defined “region of interest”
>
> The region of interest is defined to include configurations that can be potentially verified by testing and are effective in minimizing the free objective. We demonstrate in the experimental results that introducing the “region of interest” is almost always preferable compared to the baseline methods that consider the full front. In addition, we added experiments showing that the specified region is preferable over a one-sided bound (Fig. C.7) and that the method is generally insensitive to the choice of $\delta’$ that controls the region’s width (Fig C.6).
>
> >  On the balance between exploration and exploitation
>
> The overall approach does not rely on the optimization mechanism, as it suggests to perform optimization over a portion of the Pareto front in order to improve statistical efficiency while selecting a risk-controlling configuration.
> Here we implement this by defining a new reference point that encloses the region of interest. This can be coupled with several acquisition functions that are defined with respect to a reference point in the multi-objective case, such as hypervolume maximization, expected hypervolume maximization and probability of hypervolume improvement. These methods offer different exploration vs exploitation balancing possibilities. In our case, we found hypervolume maximization to be effective, where there is more emphasis on exploitation. It may be attributed to the fact that the region of interest narrows the search space focusing on a relatively small region.
>
> > On the computational overhead compared to hyperparameter optimization techniques
>
> The computational overhead of the proposed method compared to traditional hyperparameter techniques is in the testing stage. We need to evaluate the configurations in the chosen set, which requires a forward-pass of the model per tested setting. This is negligible with respect to the BO part that requires retraining the model N times.

---

> > ### Author Response · Authors · 2023-11-23
> > **Discussion**
> >
> > Once again, thank you for your time and efforts in reviewing our paper, and for your valuable feedback.
> >
> > As the discussion period is about to end soon, we would greatly appreciate it if you could go over our response and revised manuscript, and please let us know if they resolve your questions, or if there are any further concerns that need to be addressed.

---

### Author Response · Authors · 2023-11-20
**General response to all reviewers**

We would like to thank the reviewers for their time and effort in reviewing our paper. Their constructive comments helped us to improve the paper, as reflected in the revised manuscript. Following their comments, we made major changes to the paper’s writing, clarifying important aspects, and providing additional details on the experimental setup. Moreover, we added new experiments and ablation studies that strengthen the empirical verification of our method.

We would like to emphasize the contribution of our paper:

(i) This is the first time that risk control is demonstrated for tuning hyperparameters that affect the training of large models. Previous literature in the field of uncertainty quantification focused on building valid prediction sets or obtaining risk control by selecting hyperparameters that are set after the model is trained. Here we present risk control in a broader sense of model selection that can be applied to any influencing hyperparameter,  while being computationally and statistically efficient.

(ii) Our core idea is to perform a focused BO procedure that is guided by the goal of improving testing efficiency.  The method is built upon “Learn then Test” and “Pareto Testing” with the motivation of improving  their statistical efficiency by a dedicated BO method.

(iii) The paper presents diverse applications in relevant significant areas that are extensively explored by the ML community and can benefit from the ability to reliably select configurations under user-defined constraints.

(iv) The experimental results demonstrate that the method is versatile, consistent and highly-performing with respect to other risk-controlling baselines by better minimizing the free objective function across different tasks and limits.

---

### Meta-Review · Area_Chair_uzSA · 2023-12-07

**Metareview:**

The authors consider a setting wherein Bayesian optimization is used for model selection under various risk-controlling constraints (such as on algorithmic fairness, cost, etc.).

The reviewers mostly agree that this setting is of general interest (although some did question the fit for the ICLR audience in particular). Further, there was shared support for the proposed approach, at least at a high level.

However, the reviewers also noted some perceived weaknesses in the manuscript as written:

- a lack of clarity regarding some details of the proposed methodology, especially regarding the "region of interest"
- some questioned the practical applicability of the proposed methodology
- questions regarding some choices made in the empirical study

**Justification For Why Not Higher Score:**

Ultimately, the reviewers judged these weaknesses to overshadow the strengths of the contribution, and no champion for the work emerged during the discussion period.

**Justification For Why Not Lower Score:**

N/A

---

### Decision · Program_Chairs · 2024-01-16

Reject